# Zero-Shot 3D Question Answering via Hierarchical View-to-Token Transportation

**Dongsheng Wang** [1]  **Dawei Su** [1]  **Hui Huang*** [1]

## Abstract

Recently, zero-shot 3D scene understanding via 2D Vision-Language Models (VLMs) has gained increasing research interest due to their promising spatial reasoning capabilities. Typically, multiple 2D views are sampled from a 3D point cloud and fed into pre-trained VLMs to answer a given question. This paradigm highlights the critical role of input context quality and raises the challenge of retaining as many task-relevant 3D details as possible under a limited input budget. We propose `KeyVT`, a hierarchical approach for input context collection at both the view and token levels. Specifically, we combine pixel features with camera parameters and assess view importance based on both semantic content and geometric position, resulting in spatially consistent and task-relevant views. Furthermore, we address redundancy among patches across selected views by identifying representative tokens under the optimal transport (OT) framework, where view tokens and key tokens are formulated as two discrete distributions in the embedding space. These key tokens are expected to cover all view features by minimizing the OT distance. We evaluate our framework on three widely used benchmarks, demonstrating significant improvements over existing tuning-free methods and performance comparable to training-based approaches.

## 1. Introduction

After processing large-scale vision and text pairs, the Vision-Language Models (VLMs, such as GPT-4o (Hurst et al., 2024) and Qwen3-VL (Bai et al., 2025a)) have demonstrated promising capabilities in various multimodal tasks, including image/video question answering, captioning, and reasoning (Yu et al., 2025b; Zhang et al., 2025b; Bai et al., 2025a; Wu et al., 2025a; Wang et al., 2024; Su & Wang, 2026). Inspired by these achievements and the success of pure vision solutions in Tesla's full self-driving system (Tesla, 2023), recent studies have begun extending 2D VLMs to the 3D physical world by sampling multiple views from 3D point clouds. Once the view sequences are obtained, 2D VLMs can naturally reason with spatial environments, showing significant potential compared to 3D language models (Daxberger et al., 2025; Wang et al., 2025a; Wu et al., 2025b). The latter typically rely on huge high-quality 3D-text data (such as point clouds paired with detailed descriptions) to align geometric structures with language (Chen et al., 2024b; Xu et al., 2024; Deng et al., 2025b). However, such annotated 3D data remain scarce, limiting their scalability and practical applicability.

The quality of the input context, therefore, plays a central role in 3D scene understanding. Taking 3D question answering (3D-QA) as a representative example, ideal inputs should incorporate as many task-relevant 3D details as possible, such as the main objects and their surrounding environments. Early studies directly employed uniform sampling strategies to select input views. While simple, these approaches often introduce substantial noise (Zhang et al., 2025b). To enhance input quality, various key-view selection methods have been proposed, including semantic similarity-based retrieval (Liu et al., 2025; Zhang et al., 2025a), maximizing useful information (Tang et al., 2025b; Zheng et al., 2025), temporal clip selection (Sun et al., 2025a), and training additional key-view selectors (Wang et al., 2025a). Despite their success, most existing methods focus primarily on query–view relevance, which may overlook critical evidence not explicitly mentioned in the query, degrading downstream performance.

More importantly, previous works typically feed the selected key views into VLMs with the number of views fixed at a small fraction of the total (such as 8 or 16 views) due to input budget constraints. Intuitively, these key views often share semantic information and exhibit significant redundancy across regions. Removing such redundancies generally does not harm prediction performance and, in fact, frees

---

[1]College of Computer Science and Software Engineering, Shenzhen University, China. Correspondence to: Hui Huang <hhzhiyan@gmail.com>.

*Proceedings of the $43^{rd}$ International Conference on Machine Learning*, Seoul, South Korea. PMLR 306, 2026. Copyright 2026 by the author(s).

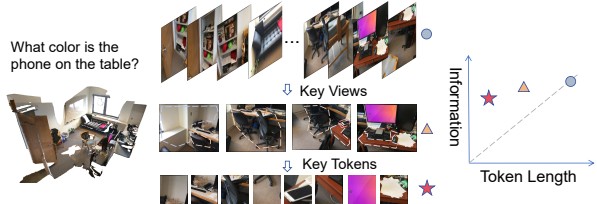

*Figure 1.* Motivation of our proposed `KeyVT`. `KeyVT` proposes to find optimal input context via a hierarchical key-view-then-key-token pipeline that preserves most information while significantly reducing the number of input tokens.

up input capacity, allowing additional key views to be incorporated without increasing computational overhead. This principle aligns with token compression techniques in video understanding tasks (Shen et al., 2025a; Wang et al., 2025b; Shang et al., 2025; Hyun et al., 2025), which typically filter out temporally redundant tokens or cluster features to extract representative information. While effective, these clustering-based models are sensitive to densely populated features and may fail to capture diverse tokens across key views (Cho et al., 2025).

To overcome these limitations and collect as much task-relevant information as possible within a predefined input budget, we propose a hierarchical framework, `KeyVT`, which first selects spatially consistent, task-relevant **Key Views** and then compresses representative **Tokens** across these views in an unsupervised manner. Moving beyond similarity-based view sampling methods, we introduce geometric parameters for each view to model spatial relationships. Our idea is simple: the main objects and their surrounding environment provide primary evidence about the corresponding regions (As shown in Fig. 1). Given the camera parameters of each view, we estimate the relative distance from the first view to the current view. This captures spatial relationships within the entire 3D world and enables segmentation into spatially consistent sub-scenes. Given the fixed number of key views, we design an adaptive strategy to identify the optimal number of selected views for each sub-scene according to its importance, *e.g.*, more relevant sub-scenes are allocated more key views.

Moving beyond the view-level selection, we further explore the token redundancy across the key views. We introduce the concept of virtual tokens, whose number is smaller than that of the key-view tokens while still being able to cover 3D details in the feature space. Unlike previous clustering-based models, we propose to learn virtual tokens under the optimal transport (OT) theory (Villani et al., 2008). OT provides a principled measure of the cost required to transport mass from one distribution to another given a predefined cost function, and recent advances have made OT computationally efficient and widely applicable (Cuturi, 2013; Li et al., 2023b; Wang et al., 2023; 2022). Specifically, all key-view tokens are represented as a redundant set that encodes

the task-relevant evidence, and the virtual token set aims to summarize key information by transporting features from the redundant set. Leveraging the ability of OT to model geometric structures in the space of probability distributions, our model is able to discover diverse and representative virtual tokens. Importantly, the learned transport plan of OT measures the probability value from virtual tokens to view tokens, which provides a simple and effective way to identify key tokens by ranking the transport plan. In summary, our contributions are three-fold:

- We propose `KeyVT`, a hierarchical framework for selecting key views and key tokens from 3D scenes, enabling 2D VLMs to access rich 3D details for effective 3D scene understanding.

- We explore the spatially consistent, task-relevant key-view selection and OT-based key token compression to jointly extract diverse and representative context under a limited input budget.

- We demonstrate that `KeyVT` achieves significant improvements on three widely used 3D-QA benchmarks, with performance comparable to fine-tuned models.

## 2. Related Works

**3D Large Language Model.** Inspired by the huge success from LLMs to VLMs (Hurst et al., 2024; Bai et al., 2025a), 3D-LLMs aim to use 3D point clouds as inputs and align such spatial features in the language space. Pioneering works such as PointLLM and LL3DA (Xu et al., 2024; Chen et al., 2024a) employ the 3D-to-text loss to learn the projection layer from points to LLMs, showing interesting results in understanding object-level points. To mitigate the lack of high-quality paired training data, subsequent efforts develop additional loss (Jin et al., 2023; Zhang et al., 2024c; Deng et al., 2025a), introduce large-scale synthetic datasets (Yang et al., 2024), and unify multiple 3D tasks under the autoregressive framework (Zhu et al., 2023; Chen et al., 2024c; Zhang et al., 2024b); Recent studies such as 3D-LLM (Hong et al., 2023b), PQ3D (Zhu et al., 2024) and BridgeQA (Mo & Liu, 2024) view image as complementary inputs and feed the multi-domain features into LLMs. Although these methods improve the 3D understanding ability, the limited training pairs hinder the alignment performance, leading to suboptimal results, especially in fine-grained 3D-QA tasks. In contrast, our approach takes 2D patches as inputs, resulting in a simpler pipeline.

**Key View Selection & Token Compression.** Due to the limited input budget, preparing the optimal input content for the VLMs is another challenge. While few works have studied key view or token selection in the 3D world. The most related work lies in the video QA tasks, as long-form videos easily exceed the visual token budgets of current video–language models (Zhang et al., 2025b; Li

et al., 2025a). To find the key frames from the video stream, learning-based algorithms aim to train various selectors (Buch et al., 2025; Yu et al., 2025a; Lee et al., 2025; Yao et al., 2025; Ghazanfari et al., 2025). Although effective for short videos, these approaches are computationally expensive and scale poorly to long-form content, motivating the development of training-free alternatives. For example, BOLT (Liu et al., 2025) improves frame diversity via inverse transform sampling, while TCoT (Arnab et al., 2025) and MDP3 (Sun et al., 2025b) leverage long-context reasoning or list-wise optimization to select candidate keyframes. AKS (Tang et al., 2025a) formulates keyframe selection as an optimization problem under fixed token budgets, while Q-Frame (Zhang et al., 2025a) introduces dynamic resolution by ranking frames into multiple granularity levels. In parallel, token compression approaches reduce visual inputs through uniform sampling or pooling (Qu et al., 2024; Wu, 2024), clustering representative frames or tokens (Shang et al., 2025; Zhang et al., 2024a; Wang et al., 2025b; 2026), or query-aware selection based on task relevance (Zhang et al., 2016; Shen et al., 2025a; Korbar et al., 2024). However, these strategies either overlook semantic importance, incur high offline costs, or rely on predefined queries, limiting their applicability to long, open-ended videos.

Focusing on 3D scene understanding, `KeyVT` differs from prior video-based approaches both in technique and application. We introduce a hierarchical selection pipeline that identifies optimal content at both the view and token levels. By leveraging camera parameters for geometry-aware key-view selection and OT-guided token compression, `KeyVT` effectively discovers spatially consistent and representative features, enabling VLMs to access richer 3D context within a limited input budget.

## 3. Method

### 3.1. Problem Formulation

In this work, we consider a 3D question answering (3D-QA) task for a 3D scene represented by a set of 2D views $\mathcal{M} = \{V_1, V_2, ..., V_{|\mathcal{M}|}\}$ and an associated question $Q$, where $|\mathcal{M}|$ denotes the number of views. Each view $V_i$ is accompanied by its camera parameters (e.g., position and orientation), which provide geometric information describing the spatial configuration of the view within the 3D scene. Due to computational and memory constraints, the available input budget $S$ (or the number of input tokens) of VLMs is typically insufficient to cover all views: $S \ll |\mathcal{M}|_t$, where $|\mathcal{M}|_t$ is the number of tokens of $\mathcal{M}$. Our goal is to find the optimal input context $\mathcal{I}$ that satisfies the input budget constraint while preserving sufficient task-relevant evidence for answering the question. Mathematically, a VLM takes $\mathcal{I}$ and question $Q$ as input and generates the answer $A$ as:

$$A = \text{VLM}(\mathcal{I}, Q), \quad \mathcal{I} = f(\mathcal{M}, Q, S), \qquad (1)$$

where $f$ is the selection function that is expected to extract useful information from all views $\mathcal{M}$ according to $Q$.

Generally, a pre-trained multimodal encoder $E$ (e.g., BLIP2 (Li et al., 2023a)) can be used to compute the cosine similarity between each view and the question. The top $K = \frac{S}{|V|}$ views with the highest similarity scores are then selected (Liu et al., 2025; Zhang et al., 2025a), where $|V|$ denotes the number of tokens per view:

$$f(\mathcal{M}, Q, S) = \{V_i | i \in \text{Top-K}(K, \cosine(E(Q), E(\mathcal{M})))\}. \qquad (2)$$

Intuitively, the higher the similarity score to the question, the higher the probability of being selected. So far, $\mathcal{I}$ is obtained solely from its visual features. However, spatial information can be crucial for understanding the 3D world. For instance, objects that are spatially closer often exhibit stronger interactions. In this study, we propose a spatially consistent key-view retrieval algorithm to update Eq. 2 by considering camera parameters.

Moving beyond view-level selection, we further investigate the redundancy among tokens (or regions) across the selected key views. Our motivation is straightforward: semantic-based retrieval strategies often select visually and semantically similar views, which in turn contain highly overlapping regions. Effectively compressing redundant tokens across views therefore becomes a critical challenge that we aim to address.

### 3.2. `KeyVT`: Key-View Key-Token Selection

In this section, we present the details of our `KeyVT` framework. As illustrated in Fig. 2, the core idea is straightforward: `KeyVT` first selects key views from the entire 3D scene based on the given question and geometric information. Then, an OT-based token compression algorithm is applied to reduce redundancy across these selected views. This hierarchical design allows VLMs to access more task-relevant 3D context within a predefined input budget, leading to improved reasoning and understanding.

**`KeyV`: Geometry-Aware Key-View Selection.** As discussed above, ideal key views are expected not only be task-relevant but also provide surrounding context, which play a core role in understanding the 3D world of the referred objects. To this end, we define a view distance $D(V_0, V_i)$ that measures both the positional and orientational differences between the first view $V_0$ and the $i$-th view $V_i$. Specifically, let $\mathbf{R} \in \mathbb{R}^{3 \times 3}$ and $\boldsymbol{t} \in \mathbb{R}^{3 \times 1}$ denote the camera orientation and position, respectively. The combined distance is computed as:

$$D(V_0, V_i) = ||\mathbf{C}_0 - \mathbf{C}_i|| + \theta(\mathbf{R}_0, \mathbf{R}_i), \qquad (3)$$

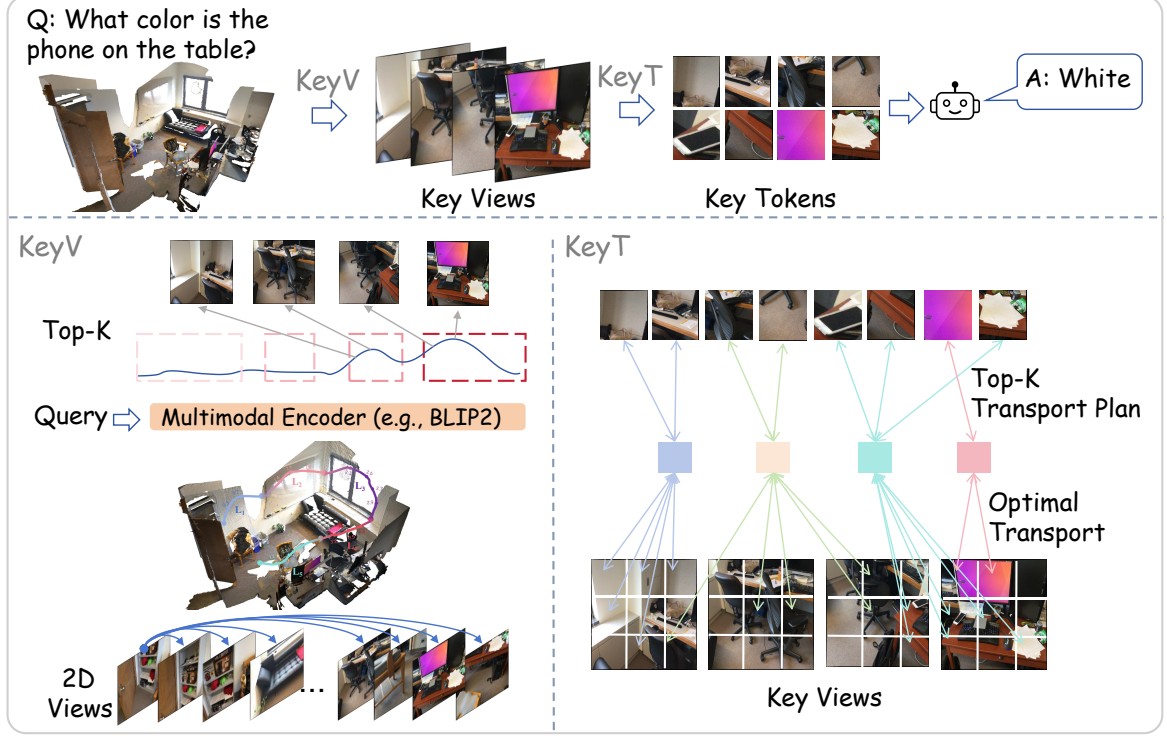

*Figure 2.* Overall framework of our proposed KeyVT. KeyVT consists of two main components: KeyV (bottom left) and KeyT (bottom right). The former introduce the geometry-aware key view selection algorithm to find spatially consistent and task-relevant views, and the latter employs the OT-guided compression to find key tokens across the key views.

where $\mathbf{C} = -\mathbf{R}^\top t$ is the camera center in world coordinates, and the angular distance $\theta(\mathbf{R}_0, \mathbf{R}_i)$ is:

$$\theta(\mathbf{R}_0, \mathbf{R}_i) = \arccos(\frac{\text{tr}(\mathbf{R}_0^T \mathbf{R}_i) - 1}{2}), \qquad (4)$$

where arccos denotes the inverse cosine function. Intuitively, the distance of Eq. 3 estimates the spatial trajectory from the first view to the $i$-th view. To segment the 3D scene into spatially consistent sub-scenes, we apply a window split over $D$ using a predefined window size $\delta$, resulting in sub-scene sequences $\{L_1, L_2, ..., L_{|L|}\}$. Note that the number of views within each $L_l$ varies, with these views capturing the specific 3D details of their corresponding sub-scene.

Not every sub-scene is equally relevant to the question. We compute the relevance score for each sub-scene $L_l$ based on its visual content:

$$r_l = \max(O_l) + \text{mean}(O_l), \quad O_{l,i \in L_l} = \text{BLIP2}(V_i, Q), \qquad (5)$$

where we employ a pre-trained multimodal encoder (such as BLIP2 (Li et al., 2023a)) to measure the semantic similarities between textual query and views within $L_l$, and the final relevant score of the sub-scene $L_l$ is obtained by combining the maximum and average scores. This design enables us to identify sub-scenes that are either consistently relevant overall or contain particularly salient views. To allocate key

views across different sub-scenes, we weight the relevance score by the size of the sub-scene:

$$N_l = \lfloor K \cdot \frac{W_l}{\sum_{l'}^{|L|} W_{l'}} \rfloor, \quad W_l = r_l \cdot \sqrt{|L_l|}, \qquad (6)$$

where $K$ denotes the total number of key views and $N_l$ is the number of views to be sampled from the $l$-th sub-scene $L_l$. Motivated by the empirical observation that larger sub-scenes tend to contain more diverse content, the weighting term $W_l$ assigns them more key views, while the square root scaling mitigates the dominance of excessively large scenes. Once $N_l$ is obtained, we can select the key views within each sub-scene $L_l$ as: $\mathcal{M}_l^* = \{V_i \mid i \in \text{Top-K}(N_l, O_l)\}$. The final set of key views is obtained by concatenating the selected views from all sub-scenes in temporal order, *i.e.*, $\mathcal{M}^* = [\mathcal{M}_1^*, \mathcal{M}_2^*, ..., \mathcal{M}_{|L|}^*]$.

**KeyT: OT-Guided Key Token Selection.** After obtaining the key views from a 3D scene, prior works typically feed the selected view set $\mathcal{M}^*$ directly into VLMs to generate answers. However, this view-level selection suffers from two limitations. First, the restricted number of key views may fail to capture sufficient fine-grained visual details. Second, it overlooks the substantial redundancy across different views, which often share overlapping regions. To this end, we move beyond view-level selection and investigate a token-level selection mechanism. Formally, let

$\mathbf{P} = \{\mathbf{e}_1, \mathbf{e}_2, \ldots, \mathbf{e}_N\}$ denote the set of token embeddings extracted from $\mathcal{M}^*$, where $\mathbf{e}_n$ represents the embedding of $n$-th token. Our goal is to construct a compact set of $M$ virtual tokens $\mathbf{Q} = \{\mathbf{c}_1, \mathbf{c}_2, \ldots, \mathbf{c}_M\}$ that preserves as much semantic information from $P$ as possible, with $M < N$.

We reformulate this task as an optimal transport (OT) problem and aim to find the optimal $\mathbf{Q}$ by minimizing the OT distance between $\mathbf{P}$ and $\mathbf{Q}$. Specifically, $\mathbf{P}$ and $\mathbf{Q}$ can be rewritten as two discrete distributions over the embedding space:

$$\mathbf{P} = \sum_{n=1}^{N} \alpha_n \boldsymbol{e}_n, \quad \mathbf{Q} = \sum_{m=1}^{M} \beta_m \boldsymbol{c}_m, \qquad (7)$$

where $\boldsymbol{\alpha}$ and $\boldsymbol{\beta}$ are simplex vectors that denote the importance of each token in $\mathbf{P}$ and $\mathbf{Q}$, respectively, and we can employ the Uniform distribution if no prior knowledge is provided. By defining a cost matrix $\mathbf{C} \in \mathbb{R}^{N \times M}$ that measures the transport cost from $\boldsymbol{e}_n$ to $\boldsymbol{c}_m$, the OT distance between $P$ and $Q$ can be expressed as:

$$d_{\mathbf{C}}(\mathbf{P}, \mathbf{Q}) = \min_{\mathbf{T} \in U(\alpha, \beta)} < \mathbf{T}, \mathbf{C} >, \qquad (8)$$

where $< \cdot, \cdot >$ denotes the Frobenius dot-product; $\mathbf{T} \in \mathbb{R}_{>0}^{N \times M}$ is the transport plan that measures the transport probability between $\boldsymbol{e}_n$ and $\boldsymbol{c}_m$; $U(\alpha, \beta) := \{\mathbf{T} \in \mathbb{R}_{>0}^{N \times M} | \mathbf{T} \mathbf{1}_M = \boldsymbol{\alpha}, \mathbf{T}^T \mathbf{1}_N = \boldsymbol{\beta}\}$; and $\mathbf{1}_N$ is the $N$ dimensional vector of ones. We specify the cost matrix $\mathbf{C}$ as the cosine distance, $e.g.$, $\mathbf{C}_{n,m} = 1 - \cos(\boldsymbol{e}_n, \boldsymbol{c}_m)$. Unfortunately, directly optimizing Eq. 8 can be time-consuming for large-scale cases, an entropy regularized OT distance is introduced in Cuturi (2013), named the Sinkhorn distance:

$$d_{\mathbf{C}, \gamma}(\mathbf{P}, \mathbf{Q}) = \min_{\mathbf{T} \in U_\gamma(\alpha, \beta)} < \mathbf{T}, \mathbf{C} >, \qquad (9)$$

where $U_\gamma(\boldsymbol{\alpha}, \boldsymbol{\beta}) = \{\mathbf{T} \in U(\boldsymbol{\alpha}, \boldsymbol{\beta}) | h(\mathbf{T}) \geq h(\boldsymbol{\alpha}) + h(\boldsymbol{\beta}) - \gamma\}$, $h(\cdot)$ is the entropy function, and $\gamma \geq 0$. This formulation yields a fully differentiable objective, enabling efficient optimization of the virtual tokens $\mathbf{Q}$ via gradient-based methods. We summarize the updating algorithm in Algorithm 1.

Once the virtual tokens $\mathbf{Q}$ are obtained, they can in principle be fed into VLMs to generate the final answer. However, VLMs are not designed to directly process these summary tokens, as they do not correspond to real image patch tokens but are instead virtual representations learned in the embedding space. To bridge this gap, we ground each virtual token using its neighboring real patch tokens. Specifically, for each virtual token $\mathbf{c}_m$, we select the $\frac{S}{M}$ patch tokens according to the learned transport plan $\mathbf{T}$:

$$\mathrm{Nei}(\boldsymbol{c}_m, \mathbf{T}) = \{\boldsymbol{e}_i | i \in \mathrm{Top\text{-}K}(\frac{S}{M}, \mathbf{T}_{\cdot, m})\}, \qquad (10)$$

where $\mathbf{T}_{\cdot, m}$ is the $m$-th column of the transport plan. The final input context $\mathcal{I}$ is then obtained by concatenating all selected tokens.

## 4. Experiments

### 4.1. Experimental Setup

**Datasets.** We evaluate `KeyVT` on three widely used 3D question answering benchmarks: ScanQA (Azuma et al., 2022), SQA3D (Ma et al., 2022), and VSI-Bench (Yang et al., 2025). Both ScanQA and SQA3D are built upon the ScanNet dataset (Dai et al., 2017), which consists of 1,513 indoor 3D scene scans. VSI-bench contains over 5,000 question-answer pairs derived from egocentric videos sourced from various datasets. We refer to Sec. A.3 for more details.

**Baselines.** We compare our `KeyVT` with recent advances, including **1)** 3D-LLMs that takes 3D or 2D+3D as inputs, such as ScanQA (Azuma et al., 2022), SQA3D (Ma et al., 2022), 3D-Vista (Zhu et al., 2023), 3D-LLm (Hong et al., 2023a), LL3DA (Chen et al., 2024b), Chat_scene (Huang et al., 2024), and 3D-LLaVA (Deng et al., 2025a); **2)** training-based models, such as cdViews (Wang et al., 2025a) and Video-3D LLM (Zheng et al., 2025); **3)** Key frame selection models, such as AKS (Tang et al., 2025a); and **4)** Token compression models, such as FLoC (Cho et al., 2025) and DivPrune (Alvar et al., 2025).

**Metrics.** We adopt widely used metrics for each benchmark. For ScanQA, we report CIDEr(Vedantam et al., 2015), BLEU(Papineni et al., 2002), METEOR(Banerjee & Lavie, 2005), ROUGE(Lin, 2004), and exact match accuracy (EM). For SQA3D, we evaluate performance using exact match accuracy (EM). For VSI-Bench, we compute mean accuracy for multiple-choice answering (MCA) tasks and relative accuracy for numerical answering (NA) tasks. We report both the overall average score and the individual scores on the eight subtasks of VSI-Bench.

**Implementation Details.** To compute semantic relevance scores between textual questions and visual views, we employ the cross-modality encoder from a pre-trained BLIP2-itm-ViT-G model (Li et al., 2023a). For the VSI-Bench that misses the geometry information of each view, we utilize FastVGGT (Shen et al., 2025b) to extract the camera parameters from each RGB frame within the 3D scenes. We set the window size $\delta$ to 1 and apply this setting across all datasets and tasks. For OT-based compression, we perform a lightweight update of the virtual tokens $\mathbf{Q}$ via Adam optimizer at a learning rate of 1e-2 within 10–15 iterations. As a plug-and-play strategy, we report that the results vary across LLaVA-OneVision(Li et al., 2024), Qwen2.5-VL(Bai et al., 2025b), and LLaVA-Video(Zhang et al., 2025b). We run all our experiments on a single 48G RTX 6000ada GPU. We refer to Sec. A.1 for more implementation details.

### 4.2. Main Results

Table 1 presents the quantitative results of various models on the ScanQA and SQA3D datasets. Since this paper fo-

*Table 1.* Evaluation Results on ScanQA and SQA3D. For compression-based methods (DivPrune, FLoC, and our KeyVT), $8^{eq}$ indicates that 16 frames are first sampled and then compressed into 8 equivalent frames for fair comparison. The best results among tuning-free models across different VLM backbones are highlighted in **bold**.

| Methods | ScanQA (val) | | | | | SQA3D (test) | input frames |
|---|---|---|---|---|---|---|---|
| | BLEU-1 | BLEU-4 | METEOR | ROUGE-L | CIDEr | EM-1 | |
| ScanQA | 30.2 | 10.1 | 13.1 | 33.3 | 64.9 | 47.2 | – |
| SQA3D | 30.5 | 11.2 | 13.5 | 34.5 | - | 46.6 | – |
| 3D-Vista | - | - | 13.9 | 35.7 | - | 48.5 | – |
| 3D-LLM | 39.3 | 12.0 | 14.5 | 35.7 | 69.4 | - | – |
| LL3DA | - | 13.5 | 15.9 | 37.3 | 76.8 | - | – |
| Chat-Scene | 43.2 | 14.3 | 18.0 | 41.6 | 87.7 | 54.6 | – |
| 3D-LLaVA | - | 17.1 | 18.4 | 43.1 | 92.6 | 54.5 | – |
| cdViews (LLaVA-OV-7B) | 42.6 | - | - | 46.8 | 94.0 | 56.9 | 9 |
| Video-3D (LLavAVideo-7B) | - | - | - | - | 96.4 | 57.0 | 8 |
| Video-3D (LLavAVideo-7B) | - | - | - | - | 100.6 | 57.8 | 16 |
| Video-3D (LLavAVideo-7B) | 47.1 | 16.2 | 19.8 | 49.0 | 102.1 | 58.6 | 32 |
| LLaVA-OV-7B | 38.5 | 10.3 | 16.4 | 43.1 | 84.0 | 53.9 | 8 |
| +AKS | 41.5 | **13.6** | 17.5 | 45.2 | 90.2 | 56.2 | 8 |
| +DivPrune | 40.3 | 13.1 | 17.3 | 45.1 | 89.5 | 53.8 | $8^{eq}$ |
| +FLoC | 41.3 | 13.7 | 17.6 | 45.6 | 91.1 | 54.8 | $8^{eq}$ |
| +KeyVT (Ours) | **41.7** | 12.4 | **17.9** | **46.7** | **93.8** | **57.1** | $8^{eq}$ |
| LLaVAVideo-7B | 43.9 | 12.2 | 18.4 | 45.9 | 93.4 | 54.7 | 8 |
| +AKS | 45.7 | 12.6 | 19.2 | 48.1 | 99.0 | 57.3 | 8 |
| +DivPrune | 46.2 | 12.3 | 19.3 | 48.5 | 99.1 | 55.8 | $8^{eq}$ |
| +FLoC | 46.4 | 10.9 | 19.3 | 48.5 | 99.4 | 57.2 | $8^{eq}$ |
| +KeyVT (Ours) | **46.6** | **14.5** | **19.4** | **48.8** | **100.7** | **57.9** | $8^{eq}$ |
| Qwen2.5VL-7B | 26.6 | 7.5 | 12.4 | 34.2 | 63.4 | 46.0 | 8 |
| +AKS | 28.3 | 7.5 | 13.2 | 36.4 | 67.2 | 49.5 | 8 |
| +DivPrune | 27.9 | 8.3 | 13.2 | 36.5 | 67.3 | 50.1 | $8^{eq}$ |
| +FLoC | 28.4 | **8.6** | 13.5 | 37.0 | 68.5 | 50.2 | $8^{eq}$ |
| +KeyVT (Ours) | **29.7** | 8.5 | **13.5** | **37.3** | **69.6** | **50.4** | $8^{eq}$ |

cuses on tuning-free approaches, we report results based on three commonly used 2D VLM backbones to evaluate the generalizability of different methods. For token compression models (e.g., DivPrune, FLoC, and our KeyVT), we first select 16 key views using the same KeyV selection strategy for a fair comparison, and then apply different algorithms to compress them into 8 key views. From the results, we have the following interesting findings. **1)** Overall, 2D-based VLMs consistently outperform existing 3D LLMs by a large margin, highlighting the strong potential of leveraging powerful 2D VLMs for 3D scene understanding tasks. **2)** Compared with both keyframe-based (AKS) and compression-based methods (DivPrune and FLoC), our approach achieves the best performance in most cases. Notably, KeyVT consistently outperforms AKS across all VLM backbones in terms of CIDEr and EM-1, two critical metrics that measure the similarity between predicted answers and ground-truth responses. This demonstrates the effectiveness of our token-level selection mechanism, which enables the model to preserve more task-relevant details under the same input budget. Moreover, the improvements

over DivPrune and FLoC validate our OT-based key token selection strategy: semantic-guided compression methods may fail to retain diverse and representative tokens, whereas KeyVT identifies optimal tokens by minimizing the optimal transport distance between virtual tokens and view tokens. **3)** Finally, when compared with tuning-based approaches such as cdViews and Video-3D, our method achieves comparable performance and even outperforms Video-3D under the same input settings. This supports the motivation behind our hierarchical selection framework—providing sufficient and informative 3D visual details is crucial for enabling 2D VLMs to effectively reason about the spatial world.

We further report results on the VSI-Bench dataset in Table. 2. Note that cdViews requires task-specific training and therefore cannot be evaluated on VSI-Bench. In contrast, tuning-free methods exhibit strong generalization to unseen 3D scenes. Table. 2 also includes See&Trek as a baseline, which explicitly incorporates motion and spatial cues for 3D understanding. Overall, KeyVT achieves the best average performance across all three VLM backbones. While

*Table 2.* Evaluation on the VSI-BENCH benchmark. All models use 8 input frames. For compression-based models (FLoC, DivPrune and `KeyVT`), 16 frames are first sampled and then compressed into 8 equivalent frames. Best resutls are highlighted in **bold**. Table. 14 shows more results on small-scale VLMs.

| Methods | Avg. | Obj. Count | Abs. Dist. | Obj. Size | Room Size | Rel. Dist. | Rel. Dir. | Route Plan | Appr. Order |
|---|---|---|---|---|---|---|---|---|---|
| | | | **Numerical Answer** | | | | **Multiple-Choice Answer** | | |
| *Proprietary Models (API)* | | | | | | | | | |
| Gemini-1.5 Flash | 42.1 | 49.8 | 30.8 | 53.5 | 54.4 | 37.7 | 41.0 | 31.5 | 37.8 |
| Gemini-1.5 Pro | 45.4 | 56.2 | 30.9 | 64.1 | 43.6 | 51.3 | 46.3 | 36.0 | 34.6 |
| GPT-4o | 34.0 | 46.2 | 5.3 | 43.8 | 38.2 | 37.0 | 41.3 | 31.5 | 28.5 |
| *Open-source Models* | | | | | | | | | |
| LLaVA-OneVision-7B | 31.4 | 34.7 | 20.6 | 47.3 | 18.1 | 40.4 | 32.4 | **32.5** | 24.9 |
| + AKS | 32.8 | 43.1 | 19.8 | 46.5 | 24.2 | **41.3** | 36.8 | 29.1 | 21.6 |
| + FLoC | 33.1 | **47.1** | **21.2** | 48.0 | 26.5 | 37.9 | 36.1 | 28.9 | 19.1 |
| + DivPrune | 33.4 | 43.2 | 21.1 | **49.4** | **28.6** | 37.9 | 37.4 | 28.5 | 21.0 |
| +See&Trek | 33.0 | 32.0 | 17.0 | 39.8 | 27.8 | 39.0 | **40.6** | 31.9 | **35.7** |
| +`KeyVT` (Ours) | **33.9** | 46.6 | 20.6 | 48.8 | 26.2 | 38.7 | 38.8 | 29.4 | 21.7 |
| LLaVAVideo-7B | 32.2 | 40.0 | 15.1 | 46.8 | 25.2 | 38.9 | 39.4 | 25.7 | 26.5 |
| + AKS | 33.2 | 41.8 | 14.6 | 48.6 | **26.8** | **41.0** | **41.3** | 33.0 | 18.8 |
| + FLoC | 34.0 | 44.1 | 22.2 | 48.7 | 25.7 | 39.4 | 39.2 | 32.5 | 20.3 |
| + DivPrune | 32.8 | 39.8 | 15.2 | **49.1** | 25.5 | 39.9 | 40.0 | 30.9 | **22.0** |
| +`KeyVT` (Ours) | **34.6** | **44.5** | **23.8** | 47.1 | 23.6 | 40.1 | 40.9 | **35.6** | 21.4 |
| Qwen2.5-VL-7B | 27.3 | 13.0 | 14.4 | 35.9 | 21.3 | **36.9** | 37.9 | 29.9 | 29.6 |
| + AKS | 36.2 | **48.0** | 23.5 | **47.0** | 28.9 | 34.1 | 37.6 | **38.1** | 32.4 |
| + FLoC | 36.6 | 46.7 | 24.8 | 45.3 | 33.2 | 36.3 | 38.5 | 35.1 | 32.1 |
| + DivPrune | 36.1 | 46.3 | 25.3 | 45.1 | 31.1 | 33.7 | 38.6 | 36.1 | 33.1 |
| +See&Trek | 29.0 | 13.6 | 14.7 | 35.4 | 23.6 | 33.4 | **41.3** | 30.4 | 39.2 |
| +`KeyVT` (Ours) | **37.0** | 41.4 | **26.7** | 46.0 | **34.8** | 31.7 | 38.7 | 35.6 | **40.8** |

*Table 3.* Performance comparison of different frame sampling strategies on the VSI-Bench, ScanQA, and SQA3D datasets. All methods sample 16 frames without additional token compression. All results are conducted on Qwen2.5-VL-7B (Qwen) and LLaVA-Video-7B (LLaVA) VLMs.

| VLMs | Method | VSI-Bench | ScanQA | | SQA3D |
|---|---|---|---|---|---|
| | | Acc | CIDEr | EM@1 | EM@1 |
| Qwen | AKS | 36.4 | 67.3 | 24.1 | 50.3 |
| | `KeyV` | **37.7** | **70.2** | **25.2** | **51.3** |
| LLaVA | AKS | 34.1 | 100.9 | 30.1 | 58.0 |
| | `KeyV` | **35.9** | **102.2** | **30.6** | **58.5** |

*Table 4.* Robustness evaluation of KeyVT across varying levels of injected camera parameter noise, alongside a performance comparison between using ground-truth parameters versus those estimated by VGGT.

| Method | ScanQA (CIDEr) |
|---|---|
| AKS | 99.0 |
| DivPrune | 99.1 |
| FLoC | 99.4 |
| KeyVT + 1% noise | 100.1 |
| KeyVT + 5% noise | 100.0 |
| KeyVT + 10% noise | 98.4 |
| True camera | 100.7 |
| VGGT camera | 100.0 |

some methods outperform `KeyVT` on individual VSI-Bench sub-tasks, they generally fail to maintain consistent performance across multiple evaluation metrics. These results demonstrate the robustness and generalizability of our hierarchical key-view–then–key-token selection strategy across diverse 3D reasoning tasks. Notably, `KeyVT` outperforms See&Trek in most settings. Although both approaches explicitly inject spatial information into VLMs, our geometry-aware partitioning strategy combined with OT-based token selection provides more effective spatial guidance.

### 4.3. Ablation Study

In this section, we analyze the impact of the two core components of `KeyVT`: `KeyV` and `KeyT`. We first compare `KeyV`

*Table 5.* Performance comparison of various KeyV components on ScanQA datasets. This set of experiments compared the results obtained by sampling 16 frames using different components and then compressing them using KeyT.

| Method | ScanQA (CIDEr) |
|---|---|
| KeyVT w/o geometry-aware design | 93.8 |
| KeyVT w/o sub-scene partition | 99.8 |
| KeyVT w/o relevance scoring | 99.4 |
| KeyVT | 100.7 |

*Table 6.* Ablation study of various keyframe selection methods integrated with the proposed KeyT. We report the CIDEr score for ScanQA,EM@1 for SQA3D, and the overall average performance for VSI-Bench.

| Method | ScanQA | SQA3D | VSI-Bench |
|---|---|---|---|
| Uniform | 93.4 | 54.7 | 32.2 |
| Uniform + KeyT | 94.0 | 55.2 | 32.7 |
| Retrieval | 97.3 | 57.4 | 32.8 |
| Retrieval + KeyT | 98.2 | 57.8 | 33.5 |
| AKS | 99.0 | 57.3 | 33.2 |
| AKS + KeyT | 99.8 | 57.7 | 33.9 |
| KeyV | 99.5 | 57.6 | 33.2 |
| KeyV + KeyT | **100.7** | **57.9** | **34.6** |

*Table 7.* Performance metrics across different key token selection models with various compression rates. The input 16 frames are given by our KeyV.

| Retain | Method | VSI-Bench Acc | ScanQA CIDEr | ScanQA EM@1 | SQA3D EM@1 |
|---|---|---|---|---|---|
| 100% | Base | 37.7 | 70.2 | 25.2 | 51.3 |
| 75% | DivPrune | 36.8 | 67.4 | 24.4 | 50.5 |
| | FLoC | 36.9 | **68.8** | **24.8** | 50.7 |
| | KeyT | **37.2** | 68.5 | 24.6 | **50.9** |
| 50% | DivPrune | 36.1 | 67.3 | 24.4 | 50.1 |
| | FLoC | 36.6 | 68.5 | 24.6 | 50.2 |
| | KeyT | **37.0** | **69.6** | **24.9** | **50.4** |
| 25% | DivPrune | 36.1 | 67.1 | 23.8 | 48.8 |
| | FLoC | 36.4 | 67.4 | 24.2 | 49.0 |
| | KeyT | **36.9** | **67.8** | **24.5** | **49.3** |

with AKS in Table. 3. As shown, KeyV consistently outperforms AKS across all three datasets. The reason is that AKS primarily relies on temporal continuity to select key views, which is well-suited for conventional video streams but less effective for 3D scenes where spatial relationships play a central role. In contrast, our KeyV explicitly incorporates geometric information to capture spatially consistent and task-relevant views. Besides, we partition KeyV into three key components and evaluate them individually in Table 5. As shown, removing any component leads to notable performance drops, with the geometry-aware design contributing the most. This validates the importance of each module KeyV. Furthermore, we evaluate the robustness of KeyV to camera parameter noise sampled from a uniform distribution, defined as $x_{\text{noise}} = x_{\text{true}} + \text{Uniform}(-x * |\epsilon|, x * |\epsilon|)$, where $\epsilon$ denotes the noise level (such as 1%, 5%, and 10%),alongside a comparison between using true camera parameters and VGGT-estimated camera parameters. As shown in Figure 4, KeyV remains robust and outperforms baselines even under 5% noise, demonstrating a strong tolerance to geometry perturbations.

In addition to KeyV, we evaluate the effectiveness of KeyT by comparing it with recent token compression methods in Table. 7. Specifically, We first report results on Qwen2.5-VL-7B using all key-view tokens as a baseline and then apply different token compression algorithms under varying compression ratios to comprehensively assess their selection quality. The result in Table. 7 demonstrates the strong compression capability of our OT-based key token selection strategy, which aims to find diverse and useful tokens by aligning the view tokens and virtual tokens in the feature space. Results on LLaVA-Video-7B are reported in Table. 15. Furthermore, we integrate KeyT into several key view selection methods. We select 16 frames with these key view selection methods and compress them with KeyT at 50% compression ratio, and compare the performance of selecting 8 frames with different key view selection methods

As shown in Table 6,the performance of obtaining an equivalent 8 frames by compressing 16 frames by 50% using KeyT is substantially better than that achieved by directly applying the key frame sampling algorithm to sample 8 frames. The combination of KeyT and our proposed method KeyV resulted in optimal performance.

### 4.4. Visualizations

Beyond the quantitative results, we provide qualitative visualizations of the key-view and key-token selection processes in Fig. 3. Given the selected key views, we project them back into the 3D scene and observe that they typically form a spatially consistent and task-relevant sub-scene, such as the regions containing the "sink" or "shelves". This behavior closely aligns with the design motivation of KeyV, which aims to preserve coherent spatial context rather than isolated views. We further project the selected key tokens back into the 3D scene. The resulting point clouds capture the principal geometric and semantic structures of the corresponding key views while effectively removing redundant regions. Notably, the 3D sub-scenes reconstructed from key tokens retain the overall structure of those reconstructed from key views, despite using significantly fewer tokens. To further assess the compression behavior, we visualize the token distributions in the embedding space using t-SNE. The results show that the selected key tokens broadly cover the distribution of the original view tokens, indicating that KeyVT preserves diverse and representative visual evidence.

### 4.5. Complexity Analysis

We further report the inference time of various methods for both the view-selection and token-selection stages in Ta-

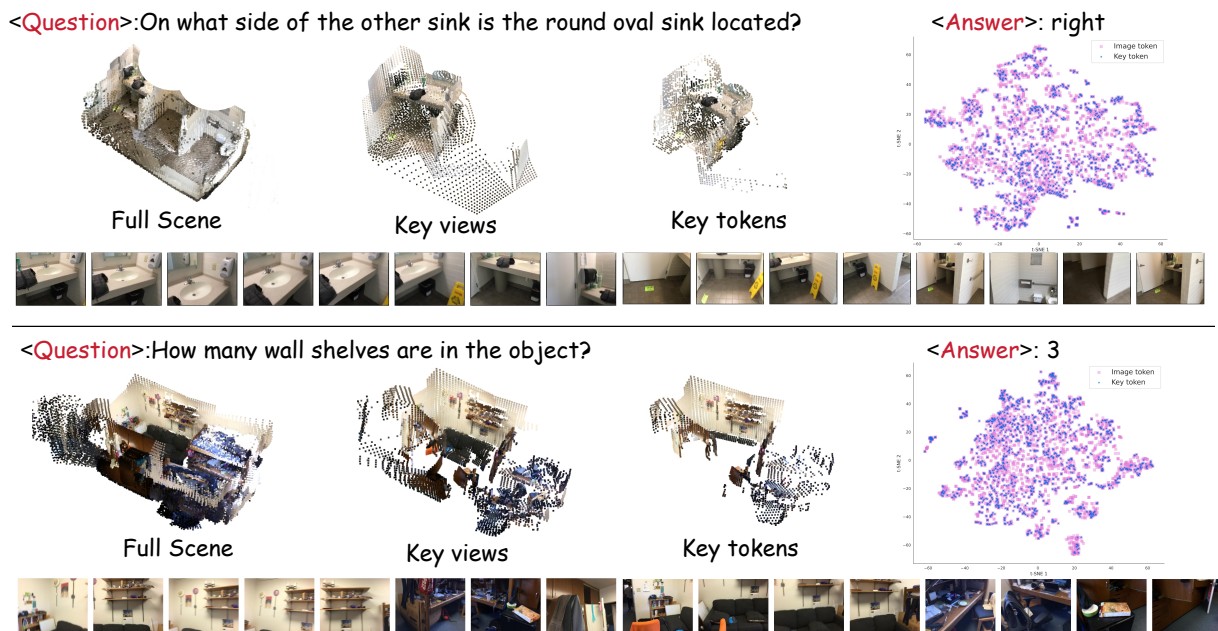

*Figure 3.* Visualization results of our key-view and key-token processes. For each case, we provide the original 3D scene, the sub-scene reconstructed from key views, and the compressed sub-scene reconstructed from key tokens. The 16 key 2D views and T-SNE results are also provided at the bottom and top-right parts, respectively. Sec. C shows more visualization results.

ble 8. Overall, our approach achieves a favorable trade-off between computational efficiency and performance. Specifically, `KeyV` exhibits inference time comparable to AKS during the key-view selection stage while incurring lower computational costs than FLoC in the token selection stage.

*Table 8.* Inference time of various models.

*(a)* Key View Selection

| Method | Time (ms) |
| --- | --- |
| AKS | 1300.79 |
| KeyV | 1302.44 |

*(b)* Key Token Selection

| Method | Time (ms) |
| --- | --- |
| Divprune | 548 |
| FLoC | 1684 |
| KeyT | 1387 |

## 5. Conclusion

We propose `KeyVT`, a hierarchical framework for key view and key token selection that enhances the input context quality of vision–language models for 3D understanding tasks. Our geometry-aware key view selection strategy jointly leverages visual features and camera parameters to identify spatially consistent and task-relevant views. Building on this, the OT-guided token compression module selects diverse and representative tokens, effectively freeing up the input budget and enabling VLMs to access richer 3D evidence. Importantly, `KeyVT` is data- and tuning-free, requiring only a few iterations of unsupervised learning to optimize virtual tokens. Extensive experiments across multiple VLM backbones and three benchmarks demonstrate the superiority of `KeyVT` in improving spatial reasoning and 3D understanding capabilities.

## Impact Statement

This work introduces `KeyVT`, a tuning-free hierarchical framework for key view and key token selection that improves the efficiency and spatial reasoning capability of existing VLMs in 3D understanding tasks. This design significantly lowers the computational and data barriers for deploying 3D-aware AI systems, potentially benefiting applications such as robotics, embodied AI, augmented reality, indoor navigation, and assistive technologies. From a broader perspective, `KeyVT` promotes more resource-efficient and accessible 3D perception, allowing strong pretrained models to generalize to unseen 3D environments with reduced reliance on large-scale labeled datasets or expensive training pipelines. This can help democratize advanced 3D understanding capabilities for researchers and practitioners with limited computational resources. The potential risks of this work are limited. Since `KeyVT` operates as a preprocessing and token-selection mechanism, it does not introduce new data sources or generate content independently. We encourage future work to explore robustness and fairness considerations when deploying `KeyVT`-enabled systems in real-world environments.

## Acknowledgments

This work was supported in part by Guangdong S&T Program (2024B0101050004), NSFC (62506237), ICFCRT (W2441020), Shenzhen Science and Technology Program (KJZD20240903100022028, KQTD20210811090044003), and Development Funds from Shenzhen University.

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

# A. Implementation Details

This section details the implementation specifics and hyperparameter configurations of the baseline algorithms. Furthermore, we provide the pseudocode for our core components (`KeyV` and `KeyT`) in Algorithm 1, along with the specific prompts utilized for each dataset as shown in Table 9. Our code is available at https://github.com/alivecat05/KeyVT.

---

**Algorithm 1** Learning algorithm of `KeyVT`.

---

**Input**: A set of $N$ 2D views $\mathcal{M} = \{V_1, V_2, ..., V_{\mathcal{M}}\}$, the question $Q$, a pre-trained BLIP2 model, the camera parameters $\mathbf{R}$ and $\boldsymbol{t}$ for each view $V_n$. The number of key views $K$, and the number of key tokens $S$. The window size $\delta$, and $M$ virtual token embeddings.
**Output**: The selected key tokens that will be fed into VLMs to generate the answer.
  // *`KeyV`: Geometry-Aware Key View selection*
Compute the relative view distance $D(V_0, V_i)$ from the first view $V_0$ to each $V_i$, $i = [1, 2, ..., |\mathcal{M}|]$, according to Eq. 3.
Obtain the sub-scenes $\{L_1, L_2, ..., L_{|L|}\}$ by apply a window split on $D$ with the window size as $\delta$.
Compute the semantic importance $r_l$ between each $L_l$ and the question via the BLIP2 model according to Eq. 5.
Identify the optimal key views for each sub-scene by weighting $r_l$ with its size $|L_l|$ according to Eq. 6.
Select $N_l$ views from sub-scene $L_l$ by applying Top-K on the question-view relevance in Eq. 5. And then concatenate all selected views to obtain the final key views $\mathcal{M}^*$.
  // *`KeyT`: OT-Guided Key Token selection*
Collect the view token set $P$ and virtual token set $Q$ according to Eq. 7.
**for** step $\leq N_{\text{step}}$ **do**
    Compute the OT distance according to Eq. 9 via the Sinkhorn algorithm.
    Update the virtual tokens $\mathbf{Q}$ via Adam optimizer by minimizing the OT distance.
**end for**
Obtain the $\frac{S}{M}$ patch tokens for each virtual token $\boldsymbol{c}_m$ according the $\frac{S}{M}$ largest transport probability in $m$-th column of the tranport plan $\mathbf{T}$.
Obtain the final key tokens by concatenating all selected tokens.

---

## A.1. Frame Sampling Baselines

To ensure a comprehensive comparison, we benchmark against two representative frame sampling algorithms:

- **AKS:** Adaptive Keyframe Sampling is a plug-and-play algorithm designed to maximize useful visual information for MLLMs by balancing text-images relevance and temporal coverage. It employs a recursive "judge-and-split" strategy that utilizes a lightweight vision-language model to score frames and adaptively partitions the video timeline to select informative keyframes. We utilize the official implementation provided by Tang et al. (2025b). Adhering to the default configuration recommended in the source code, we set the selection threshold to $s_{thr} = 0.8$ and the maximum recursion depth to $L = 4$.

- **See&Trek:** A training-free spatial prompting framework designed for vision-only MLLMs. It addresses visual homogeneity and unknown motion by employing Maximum Semantic Richness Sampling (MSRS) and integrating visual odometry-based motion cues into the input prompts. We report the results directly from the original paper (Li et al., 2025b).

## A.2. Token Compression Baselines

For token compression, we compare our method with two state-of-the-art approaches. All baselines are executed using their official codebases to ensure reproducibility.

- **DivPrune:** A diversity-based approach that selects a subset of visual tokens by maximizing the minimum pairwise distance to reduce redundancy. Following the official repository, we implement this method with a block size of 32 and utilize multiprocessing to parallelize the pruning process for efficient inference.

- **FLoC**: FLoC is a state-of-the-art, training-free token compression framework grounded in the submodular facility location function. It employs a lazy greedy algorithm to efficiently select a subset of tokens that maximizes both representativeness and diversity while minimizing redundancy. Following recommend configuration from the original paper, we adopt the temporal block processing strategy with the block length set to $T = 8$.

### A.3. Datasets

In this section, we provide a detailed introduction for all datasets that we evaluated.

- **ScanQA:** ScanQA (Azuma et al., 2022) is a 3D question answering dataset rooted in the ScanNet environments. It contains approximately 41k question-answer pairs collected from over 800 indoor scenes. The dataset distinguishes itself by requiring reasoning over the entire 3D scene geometry rather than a single viewpoint. For our evaluation, we benchmark our model on its validation set which contains 4,675 quesiton-answer pairs.

- **SQA:** SQA (Situated Question Answering in 3D Scenes) (Ma et al., 2022) focuses on embodied scene understanding. In contrast to the global perspective of ScanQA, SQA introduces a "situation" context—defined by the agent's position and orientation within a 3D scene. The dataset comprises 33.4k questions associated with 6.8k unique situations across 650 ScanNet scenes. It benchmarks a model's ability to perform situated reasoning, such as understanding spatial relations and navigation instructions from a first-person perspective. For the evaluation, we utilize the official test split, which comprises 3,519 situation-question-answer pairs.

- **VSI-bench:** VSI-bench (Visual Spatial Intelligence Benchmark) (Yang et al., 2025) is a recent benchmark designed to evaluate the fine-grained spatial reasoning capabilities of Multimodal Large Language Models (MLLMs). It consists of over 5,000 question-answer pairs derived from 288 egocentric videos sourced from ScanNet (Dai et al., 2017), ScanNet++ (Yeshwanth et al., 2023), and ARKitScenes (Baruch et al., 2021). For the experiments, We report the final average score and individual metrics on eight task types of VSI-Bench, including: (1) configurational reasoning (object counting, relative direction, absolute direction,and route planning), (2) measurement estimation (object size, rooms ize,and absolute distance),and (3) spatiotemporal reasoning (appearance order).

*Table 9.* Prompt Templates for Different Benchmarks and Question Types.

| Benchmark | Question Type | Prompt Template ($< question >$ + prompt) |
|---|---|---|
| ScanQA / SQA3D | Regression | $< question >$ + "Answer the question simply" |
| VSI-Bench | Multiple Choice | $< question >$ + "Please answer with the option's letter from the given choices (e.g., A, B, etc.) within the $< answer >< /answer >$ tags." |
| | Numerical | $< question >$ + "Please answer with the only numerical value (e.g., 42, 3.14, etc.) within the $< answer >< /answer >$ tags." |
| | Regression | $< question >$ + "Please answer with the only numerical value (e.g., 42, 3.14, etc.) within the $< answer >< /answer >$ tags." |
| | Verbal | $< question >$ + "Please answer the question simply within the $< answer >< /answer >$ tags." |

## B. Additional Ablation Studies

In this section, we present further ablation experiments on the hyperparameters of our proposed `KeyVT`.

### B.1. Hyperparameters of KeyV

**Ablation study on window size $\delta$:** As shown in Table 10, `KeyVT` is robust across a wide range of $\delta$. Specifically, an excessively small $\delta$ leads to overly fine segmentation, disrupting the spatial consistency of the sub-scenes. Conversely, as $\delta$ increases, spatially distant views are grouped together, rendering the geometry-aware partitioning ineffective and degrading the adaptive view allocation mechanism. Therefore, $\delta = 1.0$ represents the optimal setting that balances spatial consistency with effective view allocation.

*Table 10.* **Ablation study on window size.**

| Window Size | ScanQA (CIDEr) |
|:-----------:|:--------------:|
| 0.5 | 100.2 |
| 1.0 | 100.7 |
| 1.5 | 100.4 |
| 2.0 | 99.0 |
| 3.0 | 97.5 |

**Ablation on combining max and mean scores:** Furthermore, we conduct an ablation study to justify the combination of maximum (representing highly task-relevant views) and mean (representing broad scene coverage) semantic relevance scores for key view selection. As shown in Table 11, relying solely on either the maximum or mean score yields sub-optimal results. However, balancing these two factors (Combination) achieves the best performance across all three benchmarks. This demonstrates that combining both metrics effectively captures highly relevant visual evidence while maintaining sufficient environmental context, thereby improving overall 3D scene understanding and avoiding capacity waste on uninformative regions.

*Table 11.* **Ablation study on combining maximum and mean relevance scores.** We compare models using only the maximum score (highmax), only the mean score (highmean), and our proposed combination. Best results are highlighted in bold.

| Method | VSI-Bench (Avg) | ScanQA (CIDEr) | SQA3D (EM@1) |
|:-------|:---------------:|:--------------:|:------------:|
| High Max | 32.3 | 98.5 | 57.0 |
| High Mean | 32.4 | 99.0 | 57.4 |
| Combination | **34.6** | **100.7** | **57.9** |

## B.2. Hyperparameters of KeyT:

To evaluate the sensitivity of the KeyT module to optimization hyperparameters, we conduct ablation studies on iteration count and learning rate during the OT-guided virtual token update. As shown in Table 12, 10 iterations with a fixed learning rate of 1e-2 yield the best ScanQA CIDEr score (100.16), indicating that moderate optimization suffices for convergence. Table 13 shows that a learning rate of 1e-3 achieves optimal performance (100.7) under 15 iterations, while excessively large or small rates lead to instability or insufficient updates. These results confirm the robustness of our OT-based token compression across reasonable hyperparameter ranges.

*Table 12.* **Performance comparison of different iterations.**

| Learning rate | Iterations | ScanQA |
|:-------------:|:----------:|:------:|
| 1e-2 | 5 | 100.15 |
| 1e-2 | 10 | **100.16** |
| 1e-2 | 15 | 99.60 |

*Table 13.* **Performance comparison of different learning rates.**

| Learning rate | Iterations | ScanQA |
|:-------------:|:----------:|:------:|
| 1e-4 | 15 | 100.3 |
| 1e-3 | 15 | **100.7** |
| 1e-2 | 15 | 99.6 |
| 5e-2 | 15 | 100.5 |

## C. More Visualization and Failure Cases

In this section, we present additional visualization to demonstrate the effectiveness of KeyVT.

**Visualizations of Correct Cases:** As illustrated in Figure 4 and Figure 5, the visualizations of the unprojected patch point clouds provide insight into attention mechanism of KeyVT. It can be observed that KeyVT successfully identifies and extracts task-relevant sub-scenes while retaining the essential environmental context required for spatial reasoning. By effectively filtering out irrelevant background noise and focusing on the critical regions associated with the textual query, the model ensures that the reasoning process is grounded in the correct visual evidence, thereby yielding accurate answers.

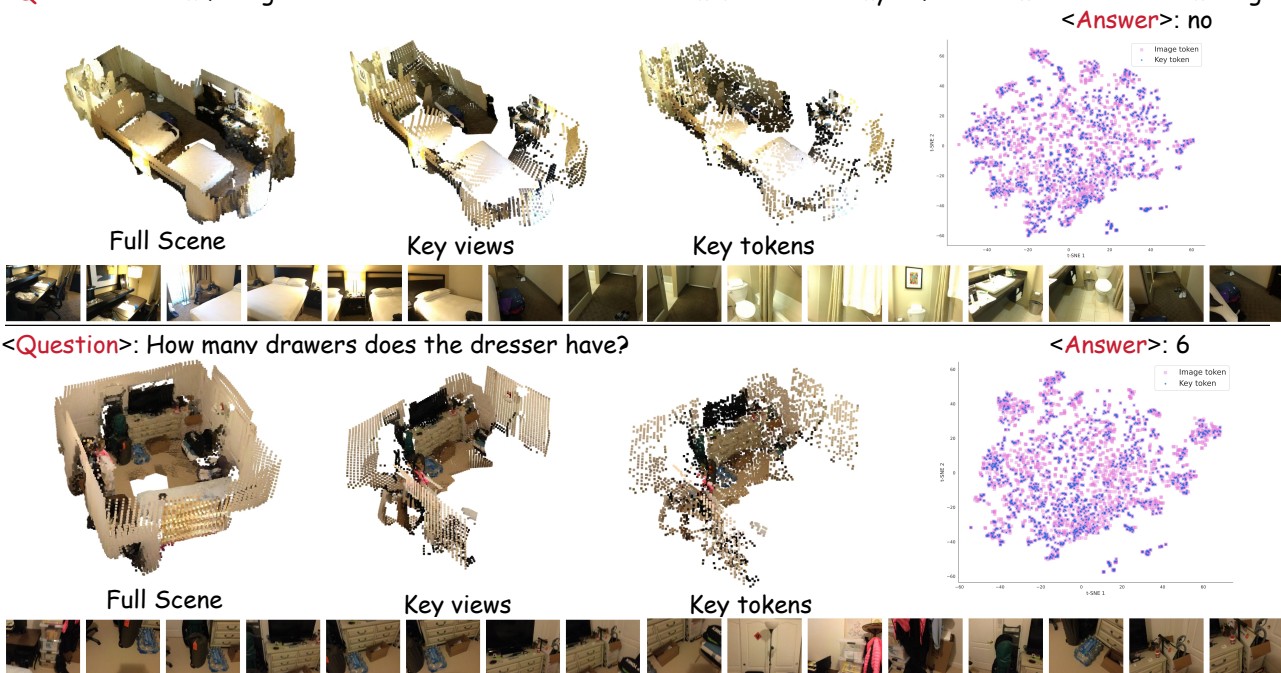

*Figure 4.* Visualization of correct cases.

**Failure Analysis:** We observe that KeyVT struggles with dense clutter or ambiguous spatial relations (Figure 6). These failures stem from a misalignment between the extracted key tokens and the textual query, which prevents the model from strictly isolating the target objects necessary for reasoning, resulting in erroneous responses.

## D. More Experiments

In this section, we provide extensive experiments on VSI-bench using small-scale models, alongside further ablation studies of token compression across different MLLM backbones.

**Small Scale Model Experiments:** As shown in Table 14, we benchmark our proposed KeyVT on small-scale models. The results demonstrate that KeyVT consistently achieves superior performance across different model scales, validating its effectiveness even under resource constraints.

**Token Compression on Various MLLM Backbones:** As shown in Table 15, we conduct extensive ablation studies on various MLLM backbones, ranging from Qwen2.5-VL-7B to LLaVA-Video-7B. The results show that our KeyT is not only practical on Qwen2.5-VL-7B but also effective on LLaVA-Video-7B across various token retention rates.

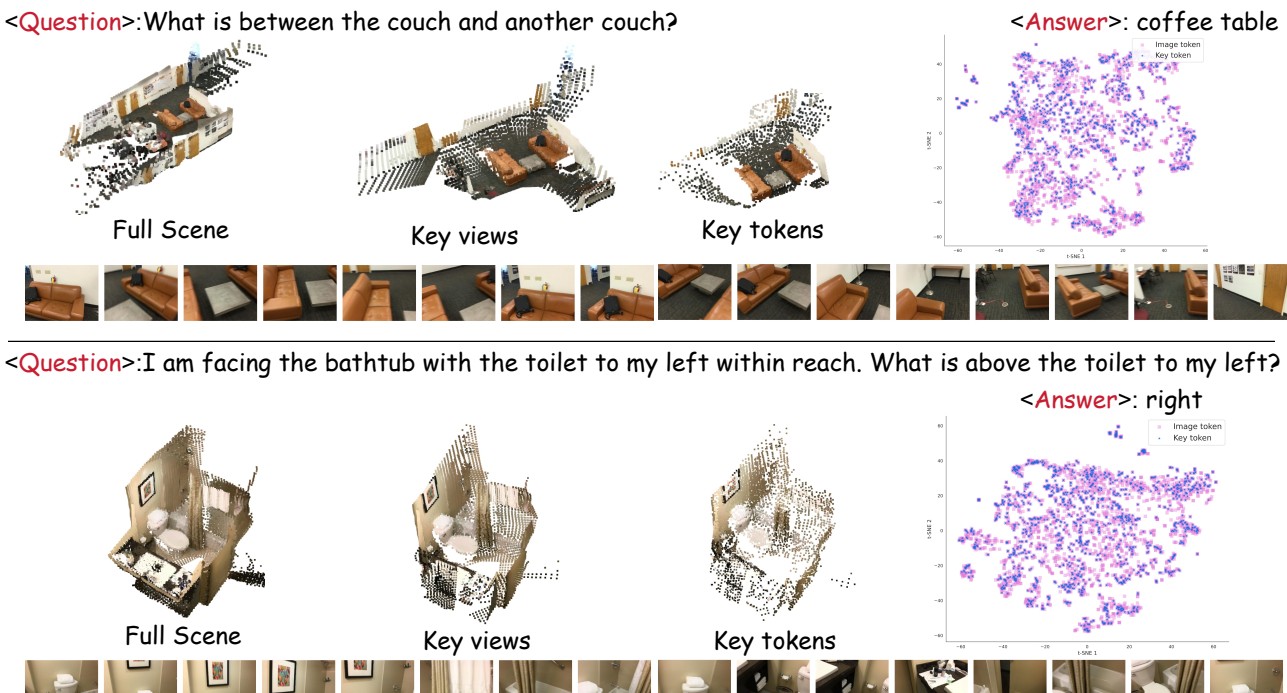

Figure 5. Visualization of more correct cases.

Table 15. Extensive ablation study on different MLLM backbone with different token compression methods. The input setting is completely same as Table 7.

| Model | Retain | Method | VSI-Bench | ScanQA | | SQA3D |
|---|---|---|---|---|---|---|
| | | | Acc | CIDEr | EM@1 | EM@1 |
| Qwen2.5-VL-7B | 100% | Full tokens | 37.7 | 70.2 | 25.2 | 51.3 |
| | 75% | DivPrune | 36.8 | 67.4 | 24.4 | 50.5 |
| | | FLoC | 36.9 | **68.8** | **24.8** | 50.7 |
| | | KeyT | **37.2** | 68.5 | 24.6 | **50.9** |
| | 50% | DivPrune | 36.1 | 67.3 | 24.4 | 50.1 |
| | | FLoC | 36.7 | 68.5 | 24.6 | 50.2 |
| | | KeyT | **37.0** | **69.6** | **24.9** | **50.4** |
| | 25% | DivPrune | 36.1 | 67.1 | 23.8 | 48.8 |
| | | FLoC | 36.4 | 67.4 | 24.2 | 49.0 |
| | | KeyT | **36.9** | **67.8** | **24.5** | **49.3** |
| LLaVA-Video-7B | 100% | Full tokens | 35.9 | 102.2 | 30.6 | 58.5 |
| | 75% | DivPrune | 34.0 | 99.7 | 30.1 | 56.4 |
| | | FLoC | 34.2 | 100.4 | 30.3 | **58.7** |
| | | KeyT | **35.0** | **101.1** | **30.4** | 58.3 |
| | 50% | DivPrune | 32.8 | 99.1 | 29.8 | 55.8 |
| | | FLoC | 34.0 | 99.4 | 30.0 | 57.2 |
| | | KeyT | **34.6** | **100.7** | **30.3** | **57.9** |
| | 25% | DivPrune | 31.7 | 88.8 | 27.5 | 54.7 |
| | | FLoC | 32.5 | 90.7 | 27.8 | 55.5 |
| | | KeyT | **33.0** | **92.9** | **28.9** | **56.4** |

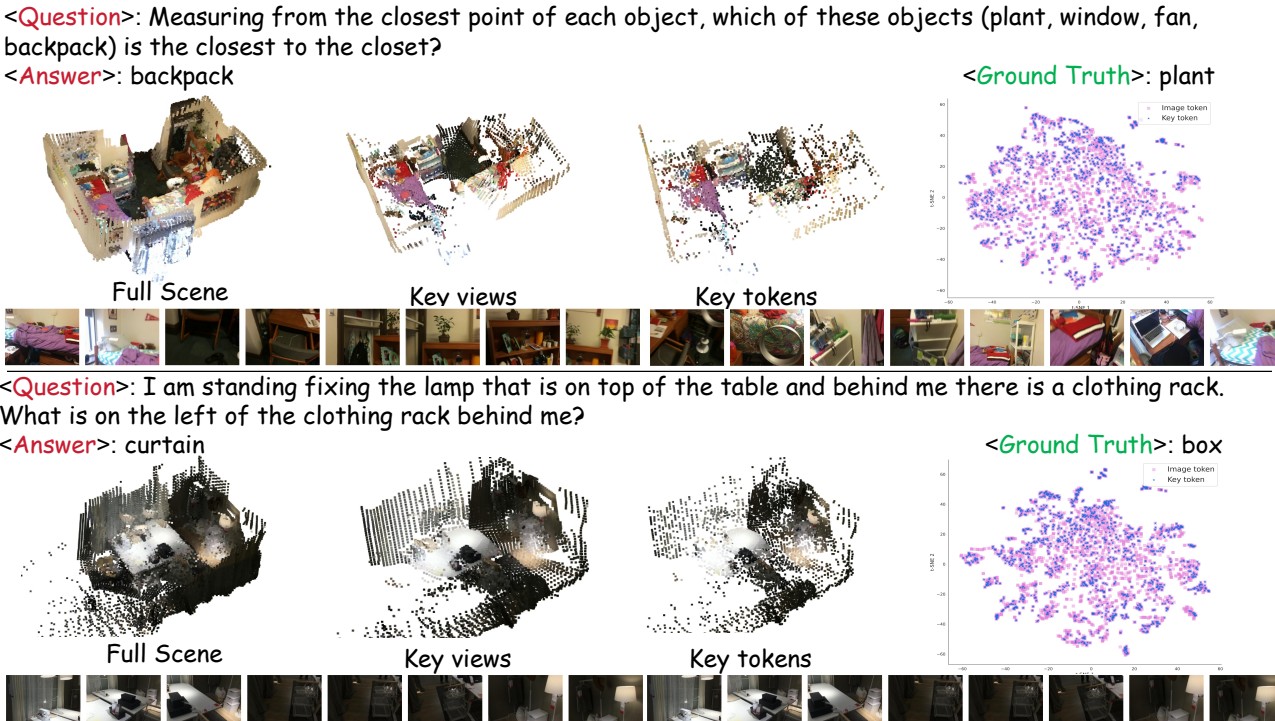

<Question>: Measuring from the closest point of each object, which of these objects (plant, window, fan, backpack) is the closest to the closet?
<Answer>: backpack          <Ground Truth>: plant

Full Scene          Key views          Key tokens

<Question>: I am standing fixing the lamp that is on top of the table and behind me there is a clothing rack. What is on the left of the clothing rack behind me?
<Answer>: curtain          <Ground Truth>: box

Full Scene          Key views          Key tokens

*Figure 6.* Visualization of wrong cases.

*Table 14.* Extensive evaluation of various baselines on VSI-bench using small-scale models.

| Methods | Avg. | Obj. Count | Abs. Dist. | Obj. Size | Room Size | Rel. Dist. | Rel. Dir. | Route Plan | Appr. Order |
|---|---|---|---|---|---|---|---|---|---|
| | | **Numerical Answer** | | | | **Multiple-Choice Answer** | | | |
| LLaVA-OneVision-0.5B | 27.1 | 33.3 | 29.2 | 13.0 | 29.1 | 29.2 | 40.2 | 36.1 | 6.6 |
| + AKS | 27.2 | 54.4 | 31.3 | 9.5 | 12.9 | 25.9 | 39.2 | 35.1 | 12.4 |
| + FLoC | 28.6 | 57.8 | 31.3 | 10.6 | 13.5 | 27.7 | 39.1 | 35.0 | 13.9 |
| + DivPrune | 27.8 | 54.2 | 31.3 | 10.3 | 13.0 | 27.9 | 38.9 | 34.5 | 12.3 |
| +See&Trek | 28.7 | 49.0 | 29.4 | 15.1 | 27.7 | 30.3 | 37.3 | 35.1 | 6.1 |
| +KeyVT | **29.2** | 38.2 | 28.5 | 17.9 | 33.0 | 30.5 | 42.5 | 35.0 | 8.0 |
| Qwen2.5-VL-3B | 25.7 | 15.0 | 17.4 | 16.0 | 27.0 | 35.1 | 44.6 | 29.9 | 21.1 |
| + AKS | 31.0 | 36.2 | 18.1 | 31.8 | 23.9 | 35.6 | 44.4 | 35.6 | 22.2 |
| + FLoC | 32.5 | 27.1 | 12.3 | 43.6 | 40.2 | 36.9 | 37.7 | 30.4 | 31.6 |
| + DivPrune | 30.1 | 35.2 | 17.2 | 32.5 | 25.3 | 33.1 | 42.9 | 32.0 | 22.7 |
| +See&Trek | 26.7 | 9.7 | 23.7 | 19.0 | 22.7 | 33.2 | 47.0 | 29.4 | 28.8 |
| +KeyVT | **33.2** | 57.1 | 26.7 | 30.9 | 18.6 | 32.4 | 45.8 | 29.4 | 24.4 |

