# OpenReview forum: "Zero-Shot 3D Question Answering via Hierarchical View-to-Token Transportation"
_ICML.cc/2026/Conference — ICML 2026 regular_

### Official Review · Reviewer_pjh6 · 2026-02-15

**Soundness:** 3
**Presentation:** 3
**Significance:** 3
**Originality:** 3
**Overall Recommendation:** 4
**Confidence:** 3

**Summary:**

This paper proposes KeyVT, a tuning-free hierarchical context selection pipeline for zero-shot 3D question answering with 2D VLMs under a tight input-token budget. It first performs geometry-aware key-view selection (KeyV) by partitioning views into spatially consistent sub-scenes using camera pose distances and allocating view budgets based on a combined semantic relevance score. It then performs OT-guided key-token selection (KeyT) to compress redundant patch tokens across selected views by learning a small set of “virtual tokens” via Sinkhorn optimal transport and using the learned transport plan to pick representative real tokens for VLM input.

**Compliance With Llm Reviewing Policy:**

Affirmed.

**Final Justification:**

My concerns have been addressed, so I will raise my score to 4.

**Key Questions For Authors:**

NA

**Limitations:**

see Weaknesses

**Strengths And Weaknesses:**

Strengths:
1. The paper is well-motivated and interesting.
2. The experimentat results are superior.

Weaknesses:
1. The contributions largely combine (i) heuristic geometry-aware view grouping/allocation with (ii) an OT/Sinkhorn-based variant of representative token selection, which is conceptually close to clustering/coverage-style compression. The paper could clarify what theoretical/empirical advantage OT provides beyond strong, simpler diversity-aware baselines.
2. KeyV module has several hyperparameters. sensitivity and failure cases are not deeply explored.
3. KeyT appears primarily distribution/coverage-driven over key-view tokens; it is not obvious that the selected tokens are question-conditioned beyond KeyV’s filtering. A direct comparison to query-aware token selection methods or a query-conditioned OT variant would strengthen the claim of preserving “task-relevant” evidence.

---

> ### Author Rebuttal · Authors · 2026-03-31
>
> We thank reviewer pjh6 for your insightful comments and suggestions! We appreciate your positive reviews that the overall idea is well motivated. Below are our point-to-point clarifications to address your concerns. Feel free to discuss with us if you have further questions!
>
> >w1: The paper could clarify what theoretical/empirical advantage OT provides beyond strong, simpler diversity-aware baselines.
>
> **Table 1. Ablations on token compression.**
>
> | Method  | VSI-Bench (Avg) | ScanQA (CIDEr) | SQA3D (EM@1) |
> |---|-----|-----|----|
> | Random Drop  | 33.6   |64.0   |  48.4   |
> | K-Means      | 35.7  | 65.4    | 49.1  |
> |DivPrune|36.1|67.3|50.1
> |FLoC |36.6|68.5|50.2
> | OT (Ours) | 37.0   | 69.6 | 50.4  |
>
> - Empirically, we provide additional results using clustering-based token compression methods in Table 1. These results show that our OT-based strategy consistently selects more informative tokens and outperforms all baselines.
> - Mathmetically, we want to clarify that the use of OT is not merely a heuristic replacement for clustering, but provides a *strictly stronger geometric formulation* for token selection.
>
> A typical clustering-based selection (e.g., k-means) can be written as:
>
> $\min_{\{c_k\}} \sum_{i} \min_{k} \|x_i - c_k\|^2$
>
> or equivalently:
>
> $\min_{a, c} \sum_{i,k} a_{ik} \|x_i - c_k\|^2
> \quad \text{s.t. } a_{ik} \in \[{0,1\}], \; \sum_k a_{ik}=1$
>
> In contrast, OT solves:
>
> $\min_{\gamma \in \Pi(p,q)} \sum_{i,j} \gamma_{ij} \, c(x_i, y_j)$, $\text{s.t. } \sum_j \gamma_{ij} = p_i,\quad \sum_i \gamma_{ij} = q_j$
>
> Clustering corresponds to a discrete, hard-assignment restriction of OT: $\( \gamma_{ij} \in [\{0, p_i\}] \)$. Therefore, the feasible set of clustering $ \mathcal{C} $ is a strict subset of the OT feasible set $ \mathcal{T} $:
> $\mathcal{C} \subset \mathcal{T}$.
>
> Unlike local distortion objective in clustering, OT introduces **cross-sample coupling** through $ \gamma $, meaning that assignments are jointly optimized across all points. This leads to better preservation of the **global geometry of the feature distribution**, rather than independent nearest-center assignments.
>
> More importantly, The OT objective induces the Wasserstein distance [1]:
> $W(p,q) = \min_{\gamma \in \Pi(p,q)} \sum_{i,j} \gamma_{ij} \|x_i - y_j\|$. A key property is that Wasserstein distance respects the underlying geometry of the space. For example, for a small shift $ \epsilon $:
> $W(p, p(\cdot - \epsilon)) \approx \epsilon
> $.
> OT captures *spatial structure* and *support mismatch*, which is critical for geometry-aware token selection.
>
> [1] Soheil et.al., Generalized Sliced Wasserstein Distances. In NeurIPS2019.
>
> >w2: KeyV module has several hyperparameters.
>
> **Table2:  Ablation study on 'window_size'**
> | Window_size | ScanQA(CIDER) |
> |---|---|
> | 0.5 | 100.2 |
> | 1 | 100.7 |
> |1.5| 100.4
> | 2 | 99.0 |
> | 3 | 97.5 |
>
> We clarify that KeyV has only one hyperparameter, the window size $\delta$. all other parameters in Eq. (5–6) are computed automatically. As shown above, KeyVT is robust across a wide range of $\delta$.
>
> >w3: KeyT appears primarily distribution/coverage-driven over key-view token.
>
> We emphasize that query-aware selection and token compression are decoupled in KeyVT. KeyV selects query-relevant views using BLIP2 scores, while KeyT focuses on removing redundant tokens across these already query-aware views.
>
> **Table3. Results of various token compressions**
>
> | Method   | ScanQA (CIDEr) | SQA3D (EM@1) | VSI-Bench (Avg) |
> |----------|----------------|--------------|-----------------|
> | PruneVid | 95.8           | 55.2         | 33.4            |
> | KeyT     | 100.7          | 57.6         | 34.6            |
>
> Following the suggestion, we include results of PruneVid [1], a query-conditioned token pruning method. Our results show that decoupling query-aware selection from token compression leads to significantly better performance.
>
>
> [1] Huang et.al., PRUNEVID: VISUAL TOKEN PRUNING FOR EFFICIENT VIDEO LARGE LANGUAGE MODELS.

---

> > ### Author Rebuttal · Reviewer_pjh6 · 2026-04-03
> >
> > The rebuttal improves the paper by adding useful ablations and clarifying the intended division of roles between KeyV and KeyT. However, my concerns are only partially resolved, especially regarding the distinct advantage of OT beyond simpler diversity-aware compression and the extent to which KeyT is truly task-relevant rather than mainly coverage-driven. I will maintain my previous score.

---

> > > ### Author Response · Authors · 2026-04-03
> > >
> > > We thank the reviewer for the feedback and for acknowledging the improvements in ablations and clarification of KeyV/KeyT roles. We address your follow-up concerns below.
> > >
> > > > The distinct advantage of OT
> > >
> > > We agree that both OT-based selection and diversity-aware methods (e.g., clustering or coverage maximization) aim to reduce redundancy. However, the key distinction is that: **clustering answers “where are the centers?”, while OT answers “how should mass be transported?”.**
> > >
> > > - Most clustering algorithms rely on minimizing point-to-center distances, which implicitly favors high-density or dominant modes in the feature space. As a result, smaller or less frequent semantic patterns can be underrepresented or even ignored, since they contribute less to the overall objective.
> > > - In contrast, the OT formulation operates at the distribution level and aims to achieve distribution-to-distribution alignment:
> > >
> > > $\quad \quad$ $P=\sum_{i=1}^N p_i \delta_{x_i}$, $\quad$ $Q=\sum_{j=1}^M q_j \delta_{y_j}$,
> > >
> > > $\quad \quad$  $OT(P,Q) = \min_{\gamma \in \Pi(p,q)} \sum_{i,j} \gamma_{ij} \, c(x_i, y_j)$, $\text{s.t. } \sum_j \gamma_{ij} = p_i,\quad \sum_i \gamma_{ij} = q_j$.
> > >
> > > In KeyVT, we specify $p$ and $q$ as uniform distributions, treating each point in $P$ and $Q$ equally. Due to the constraints in the learned transport plan, i.e., $\sum_j \gamma_{ij} = p_i$ and $\sum_i \gamma_{ij} = q_j$, each point $x_i$ is matched to targets $y_j$ not only based on semantic similarity, but also in consideration of assignments made by other points. If a target $y_j$ has already received sufficient mass from other points, $x_i$ is encouraged to match with alternative targets. This results in a coupled assignment problem, where matching decisions are not independent for each token, but are jointly constrained across all tokens. Consequently, OT prevents the solution from collapsing onto a few dominant regions and better preserves the structure of the data manifold.
> > > - We can also understand the advantage of OT from the perspective of the Wasserstein distance. The OT distance is explicitly defined over the underlying metric space, as it directly incorporates the ground cost $c(x_i, y_j)$ when measuring discrepancies between distributions. As a result, differences between distributions are quantified based on how far mass must be transported in the space, making the metric inherently sensitive to the geometric structure of the data. Therefore, OT tends to preserve the geometry of the data manifold, whereas clustering methods only capture a set of representative locations.
> > > - Compared to recent diversity-aware methods such as FLoC, which aim to avoid dominant modes via segmentation, our approach is also more efficient. FLoC requires complex iterative procedures to identify diverse centers, leading to higher computational cost (Table 5 in the submission). Empirically, our KeyVT outperforms FLoC in most cases (see Table 4 in the submission).
> > >
> > > > The extent to which KeyT is truly task-relevant rather than mainly coverage-driven
> > >
> > > - We would like to clarify again that, in KeyVT, query-aware selection and token compression are decoupled. The task-relevant sub-scenes are first identified by KeyV, while the role of KeyT is to reduce redundancy within these selected regions. In other words, KeyT focuses on structural coverage of the KeyV outputs, rather than being directly conditioned on the query.
> > > - Furthermore, we would like to emphasize that enforcing query-awareness in KeyT may not necessarily improve spatial understanding. In 3D reasoning tasks, the surrounding environment often provides critical contextual information. Selecting only query-relevant tokens can sometimes be misleading, especially in cases where the answer is not explicitly mentioned in the query.
> > >
> > > We hope our response has addressed your concerns. Please do not hesitate to reach out if you have any further questions.

---

### Official Review · Reviewer_GzrC · 2026-03-10

**Soundness:** 2
**Presentation:** 2
**Significance:** 2
**Originality:** 2
**Overall Recommendation:** 4
**Confidence:** 5

**Summary:**

This paper studies zero-shot 3D question answering with 2D vision-language models under limited visual token budgets. It proposes **KeyVT**, a hierarchical context selection framework that first performs geometry-aware key-view selection (**KeyV**) and then applies optimal-transport-based key-token compression (**KeyT**) to preserve informative 3D evidence under the same input budget. Experiments on ScanQA, SQA3D, and VSI-Bench show that KeyVT consistently improves over prior tuning-free baselines and is competitive with some training-based methods.

**Compliance With Llm Reviewing Policy:**

Affirmed.

**Final Justification:**

The rebuttal provides clear explanations and supporting experiments, so I raise my score from 3 to 4.

**Key Questions For Authors:**

1. Given that KeyVT appears to be a fairly general context-selection framework for video-based 3D scene understanding, can the authors discuss or provide evidence of its applicability beyond 3D-QA, e.g., to tasks such as 3D visual grounding?

2. Can the authors better disentangle the respective contributions of geometry-aware design, sub-scene partitioning, and relevance scoring in KeyV?

3. How scalable is KeyV in practical online settings, given that it requires scoring a large number of candidate views with a multimodal encoder such as BLIP2?

4. Can the authors provide stronger comparisons to show that the OT-based KeyT design is preferable to simpler token selection or compression baselines?

5. How sensitive is the method to inaccurate camera parameters, especially on VSI-Bench where poses are extracted using an external model?

**Limitations:**

yes

**Strengths And Weaknesses:**

***Strengths***

1. The paper addresses a practical problem in zero-shot 3D-QA, namely how to preserve informative 3D evidence under a strict VLM input budget.

2. The proposed hierarchical design is intuitive and well aligned with the problem: KeyV selects spatially coherent and question-relevant views, while KeyT further removes redundancy at the token level.

3. The method is training-free and shows consistent gains over prior baselines across multiple benchmarks and backbones.

***Weaknesses***
1. The overall contribution is more of a carefully engineered context-selection pipeline  than a fundamentally new 3D reasoning paradigm, so the methodological novelty is somewhat limited.

2. The paper does not fully disentangle whether the gains in KeyV come from the geometry-aware design itself, the sub-scene partition strategy, or the relevance scoring mechanism.

3. KeyV requires computing question-view relevance scores over a potentially large number of candidate views, using multimodal encoder like BLIP2. This may introduce substantial preprocessing overhead and limit scalability in practical real-time or long-horizon settings.

4. The benefit of KeyT is not compared in depth against simpler token compression or selection strategies, so it remains unclear whether the OT-based design is truly necessary.

5. The method relies on camera parameters, but the sensitivity to inaccurate estimated poses is not thoroughly analyzed, especially on VSI-Bench where poses are obtained from an external model.

---

> ### Author Rebuttal · Authors · 2026-03-31
>
> We thank Reviewer GzrC for your insightful comments and suggestions! We appreciate your positive reviews on the consistent performance and hierarchical design. Below are our point-to-point clarifications to address your concerns. Feel free to discuss with us if you have further questions!
>
> >W1: The overall contribution is somewhat limited.
>
> - First, we emphasize that token efficiency is a critical challenge in MLLMs, especially for 3D scene understanding, where reasoning requires substantial visual evidence. In this work, we propose a tuning-free, hierarchical framework that improves token selection and enhances 3D reasoning performance. We disagree that our novelty is limited mainly because we focus on the context-selection pipeline rather than a new 3D reasoning paradigm.
> - Secondly, our baselines (e.g., cdViews, AKS, DivPrune, FLoC) are all published in top-tier venues (e.g., CVPR, ICML, ICLR) and similarly focus on view selection or token compression without redefining the reasoning paradigm, highlighting the importance of this research direction.
> - More importantly, our hierarchical design provides key insights: (1) KeyV selects optimal views by jointly considering 3D geometry and query relevance, and (2) KeyT introduces an OT-based token compression strategy to reduce redundancy across views, enabling more informative inputs under a fixed budget.
>
> >W2 & Q2: Ablations on KeyV.
>
> **Table1. Ablations of KeyV**
>
> | Method                  | ScanQA (CIDEr) |
> |-----|-----|
> | KeyVT w/o geometry-aware design      | 93.8 |
> | KeyVT w/o sub-scene partition     | 99.8 |
> | KeyVT w/o relevance scoring      | 99.4 |
> |KeyVT |100.7|
>
> Following the suggestion, we report KeyV ablations in Table 1. Removing any component leads to notable performance drops, with geometry-aware design contributing the most. This validates the importance of each module in our framework.
>
> >W3 & Q3: KeyV requires computing question-view relevance scores.
> We acknowledge the scalability concern, particularly in real-time settings. However, this limitation is common in retrieval-based methods (e.g., AKS, cdViews), as improved relevance modeling inevitably incurs additional computational cost. As the saying goes, there is no free lunch. For example, BLIP2 enables more accurate selection of query-relevant views, but it comes with increased computational overhead. One of the main motivations of this work is to design a hierarchical framework that helps MLLMs select optimal inputs under a fixed budget. we will leave the scalability as our future work. Potential directions include: (1) applying interpolation strategies to avoid computing BLIP scores for every frame, and (2) distilling a lightweight version of BLIP to reduce computational cost.
>
> >W4 & Q4: whether the OT-based design is truly necessary.
>
> **Table2. Ablations on token compression**
>
> | Method  | VSI-Bench (Avg) | ScanQA (CIDEr) | SQA3D (EM@1) |
> |---|-----|-----|----|
> | Random Drop  | 33.6   |64.0   |  48.4   |
> | K-Means      | 35.7  | 65.4    | 49.1  |
> |DivPrune|36.1|67.3|50.1
> |FLoC |36.6|68.5|50.2
> | OT (Ours) | 37.0   | 69.6 | 50.4  |
>
> We compare different token compression strategies in Table 2. The results show that our OT-based method consistently outperforms alternatives (e.g., Random, K-Means, DivPrune, FLoC), demonstrating its effectiveness in reducing redundancy while preserving informative features.
>
> >W5 & Q5: Sensitivity analysis on camera parameters.
>
> **Table 3. Sensitivity Analysis of Camera Parameters.**
>
> | Method           | ScanQA (CIDEr) |
> |------------------|----------------|
> |AKS | 99.0|
> |DivPrune|99.1|
> |FLoC|99.4|
> | KeyVT + 1% noise  | 100.1 |
> | KeyVT + 5% noise  | 100.0  |
> | KeyVT + 10% noise | 98.4 |
> |------|---|
> | True camera      | 100.7          |
> | VGGT camera      | 100.0          |
>
> As shown in Table 3, KeyVT consistently outperforms baselines even under 5% noise, demonstrating strong robustness to camera geometry perturbations. These results further suggest that performance can be improved with more accurate camera estimation.
>
> >Q1: tasks such as 3D visual grounding.
>
> Yes, as noted, KeyVT is a general approach for video-based systems. Following your suggestion, we apply KeyVT to Video-3D LLM [1] and report results on the Multi3DRefer dataset, which evaluates visual grounding performance. The results clearly show that KeyVT improves the baseline in 3D visual grounding.
>
> **Table 4. Results on the visual grounding datasets**
>
> | Method             | Multi3DRefer (F1@0.25) | Multi3DRefer (F1@0.5)|
> |--------------------|--------------|------------|
> | Video3DLLM         | 58.0         | 52.7       |
> | Video3DLLM + KeyVT  |58.9  | 53.8   |
>
>
> [1] Zheng et.al., Video-3D LLM: Learning Position-Aware Video Representation for 3D Scene Understanding. In CVPR 2025.

---

> > ### Author Rebuttal · Reviewer_GzrC · 2026-04-04
> >
> > Thanks for the rebuttal. My concerns have been adequately addressed, so I will raise my score to 4.

---

### Official Review · Reviewer_LddP · 2026-03-11

**Soundness:** 3
**Presentation:** 3
**Significance:** 3
**Originality:** 3
**Overall Recommendation:** 5
**Confidence:** 3

**Summary:**

The authors present a hierarchical approach to compress 3D visual input data into a representative set of tokens that can easily be interpreted by powerful 2D models, optimizing for ‘completeness’ of required query-aligned information while limiting redundancy in the resulting patch-token set. Given that the approach uses a distance metric to select views and optimal transport for compression, it does not require tuning of the used (foundation) model.

**Compliance With Llm Reviewing Policy:**

Affirmed.

**Final Justification:**

The authors have provided strong rebuttals to all reviewers, and my concerns have been addressed;
 Raised my score accordingly (4 $\rightarrow$ 5)

**Key Questions For Authors:**

**[Q1]: Sub-Scene relevance score & weighting**
Assuming the case where the ‘max’ term of the relevance score is dominant, while the rest of the scene is comparably uninformative but still large: I.e. only 1-2 views out of many in the big scene really matter;
  $\textrightarrow$ This would still up-weight and therefore create a large number of key-views for this scene, although most of them don’t contain valuable information; hence, wasting capacity that could be otherwise allocated
  $\textrightarrow$ I’m curious to get the authors’ view on whether they have encountered such scenarios, and whether it ‘matters’ in practice? While more complex, it seems that considering ‘high mean’ vs ‘high max’ independently could provide further benefits (at least in theory)?

**[Q2]: Ablation**
- The authors state that the ‘window size delta’ is set to 1; What’s the impact of this choice? How robust is the method to other choices, and how do results change? Is there a trade-off or sweet spot?
- OT compression: How sensitive is the optimization to the choice of learning rate and number of iterations? Also: If only 10 iterations are used, Adam might not actually be of much benefit, or even hurt optimization (at least with default hyper-params).


**[Q3]: Insights on Robustness: Camera Positions**
I’d be curious to hear how accurate the camera positions need to be, and how dependent the authors’ approach is on this. Given that one of the experiments uses estimated camera poses, I’d be interested whether this yields any noticeable issues as error propagate and likely impact key-view selection.
  $\textrightarrow$ Could be an interesting analysis for the future to provide insight into the methods robustness to noise on the camera poses, and check how the view-selection and down-stream key-token building changes!

**[Q4]: Title**
Why “Zero-Shot” in the title? I understand that the method doesn’t require tuning of the backbone; but “zero-shot” is usually associated with ‘no previous data samples/example pairs’, which I don’t really see related to the task at hand?

**Limitations:**

None that I could find;
I highly suggest the authors include a proper section on the current limitations into future versions of the manuscript!

**Strengths And Weaknesses:**

### Strengths
**Soundness:**
- Method is well-defined, with all important equations/definition provided
- Appropriate number of experiments across different VLM backbones and datasets to show effectiveness of KeyVT, and the two core components/steps (key-View and key-Token selection) are well contrasted to alternative methods / component choices
- Pretty good set of ablations in terms of individual components’ contributions, as well as timing information and behavior across different allocated key-token capacities

**Presentation:**
- Contributions and underlying motivation clearly outlined in the introduction, and well-placed within the wider area of research;
- The paper is well written and easy to read and follow, and contains a good set of visualizations that make the core idea of the method quite easy to understand;  Further details found in the appendix, but the paper provides a set of great visualizations paired some tables to provide sufficient insights

**Originality & Significance:**
- The method is neat, effective and versatile enough to be applied to a variety of different VLMs, which supports its significance (also beyond current models)
- The authors’ choice to optimize the compression of the input data by encouraging diversity/completeness of crucial query-relevant data while reducing redundancy allows the user to ‘set’ a fixed budget of tokens; significantly raising the applicability of this method even in compute-limited settings

---
---
### Weaknesses
- Several inconsistencies in ‘naming’ across method and components:
  - The authors appear to repeatedly change between ‘virtual tokens’ and ‘key tokens’ throughout the paper, which makes it hard to follow at times;
  If ‘virtual’ is intended to define the created and ‘key’ the NN patch token, then this should be defined clearly somewhere
  - Table 3: method ‘KeyVT’ is composed of ‘KeyV’ and ‘KeyT’ – yet, Table 3 uses ‘KeyF’ as method description.
- Some overclaiming in experiment interpretation: Table 1 & l.310: “[…] 2D-vased VLMs consistently outperform existing 3D LLMs by a large margin”;
  $\textrightarrow$ Chat-Scene appears to be a 3D method if I’m not mistaken, and clearly outperforms several of the 2D LLM approaches.
- Further overclaiming in efficiency:
  Intro states “[…] allowing additional key views to be incorporated without increasing the computational overhead.” While that’s true from a ‘complexity induced by the length of the token sequence’ perspective, it doesn’t hold from the actual methods’ view:
  $\textrightarrow$ Table 5 and related description show that while KeyV is comparable to AKS and KeyT is in the range of FLoC (a bit over double Divprune), it is not explicitly stated that KeyVT obviously requires both; which does indeed result in higher overhead than any of the other methods.
  $\textrightarrow$ Note: This increase might very well be justified by the better performance, but should be clearly stated somewhere in the complexity analysis in 4.5 to be entirely upfront to the reader and avoid misunderstandings.
- Some jumps between 3D and ‘temporal’ data:
  $\textrightarrow$  Parts of the related work could be better contrasted to the authors’ own approach: e.g. l.127 “limiting their applicability to long, open-ended videos”; That might be true, but relationship between 3D pointclouds and video has not been discussed, so it seems not really directly related to your approach; Might be better to directly relate (and explain) that video-based methods can also be seen as creating 3D environments over time; hence temporal methods are related in this way, etc;
  $\textrightarrow$ Likewise, l.202 (right) defines the final set of key views as “concatenated in temporal order”, while the narrative before is mainly around 3D scenes and data;


**Minor:**
A range of additional typos and grammatical oddities throughout that could be improved, e.g.
- l.086 (right) “develop additional loss $\textrightarrow$ either “losses” or “an additional loss”;
- l.092 (right) “view image as complementary” $\textrightarrow$ either “images” or “the image”;
- l.101 (right) “While few works…” $\textrightarrow$ unclear sentence structure or 2nd half missing;
- spaces after references in Implementation Details section (VLMs), Metrics;
- etc. …

---
---
**TLDR:** I think the authors’ way of approaching this problem (mainly using the OT-based compression paired with a slightly improved initial key-view selection) is an original and neat idea; and well-supported by the experimental results.
I’m happy to consider raising my score further once the questions and weaknesses have been addressed;

---

> ### Author Rebuttal · Authors · 2026-03-31
>
> We thank Reviewer LddP for your review and valuable suggestions! We appreciate your positive assessment that the overall idea is well motivated. Below, we address each of your suggestions in detail.
> >W1: Several inconsistencies in ‘naming’ across method and components:
>
> **Virtual vs. Key Tokens:** We will add a clear definition early in Section 3.2. Specifically, virtual tokens refer to abstract distributions learned in the embedding space via Optimal Transport, while key tokens denote the actual 2D image patch tokens obtained by grounding these virtual tokens back to the input views. We will carefully revise the manuscript to ensure consistent usage of these terms.
>
> **Table 3 Typo:** We will correct “KeyF” to “KeyV” in the revision.
>
> >W2: Some overclaiming in experiment interpretation
>
> We appreciate this insightful comment. Our statement regarding 2D-based VLMs was overly broad and did not adequately acknowledge the strength of advanced 3D models such as Chat-Scene. We will revise the text accordingly.
>
> >W3: Further overclaiming in efficiency:
>
> We thank the reviewer for the suggestion. While KeyVT introduces a marginal increase in preprocessing time (Table 5), it is a strategic trade-off for improved accuracy while keeping VLM inference costs constant. We will refine the Introduction and Section 4.5 to explicitly distinguish between inference overhead and preprocessing costs, providing a more balanced discussion of this trade-off.
>
> >W4: Some jumps between 3D and ‘temporal’ data
>
> Our 3D scenes are derived from human-captured depth camera sequences, they remain intrinsically linked to video data. Prior research, such as *cdViews*, has already demonstrated that optimizing input frames via trained samplers enhances 3D scene understanding—a motivation that aligns directly with video understanding tasks. We will revise the Related Work section to explicitly clarify this relationship and eliminate ambiguity.
>
> >Q1:Sub-Scene relevance score & weighting
>
> We conduct an ablation study to justify combining max (textual relevance) and mean (scene coverage). As shown in Table 1, balancing these two factors yields the best performance, effectively improving 3D scene understanding while avoiding capacity waste on uninformative regions.
>
> **Table1: Ablation study of highmean and highmax**
> | Method | VSIBENCH(avg) | ScanQA(cider) | SQA3D(em@1) |
> |---|---|---|---|
> | highmax | 32.3 | 98.5 | 57.0 |
> | highmean | 32.4 | 99.0 | 57.4 |
> | combination | 34.6 | 100.7 | 57.9 |
>
>
> >Q2: Ablation study on 'window_size' and hyperparameters of KeyT
>
> These results show that KeyVT is robust to the window size and remains stable across a wide range of optimization settings.
>
> **Table2:  Ablation study on 'window_size'**
> | Window_size | ScanQA(CIDER) |
> |---|---|
> | 0.5 | 100.2 |
> | 1 | 100.7 |
> |1.5|100.4|
> | 2 | 99.0 |
> | 3 | 97.5 |
>
> **Table 3: Performance comparison of different iterations.**
> | Learning rate | iterations | ScanQA |
> |---|---|---|
> | 1e-2 | 5 | 100.15 |
> | 1e-2 | 10 | 100.16 |
> | 1e-2 | 15 | 99.6 | | 1e-2 | 20 | 99.3 |
>
> **Table 4: Performance comparison of different learning rates.**
> | Learning rate | iterations | ScanQA |
> |---|---|---|
> | 1e-4 | 15 | 100.3 |
> | 1e-3 | 15 | 100.7 |
> | 1e-2 | 15 | 99.6 |
> | 5e-2 | 15 | 100.5 |
>
>
>
> >Q3: Insights on Robustness: Camera Positions
>
> As shown in Table 5, KeyVT consistently outperforms baselines under varying levels of geometry noise (up to 5%), demonstrating strong robustness to camera perturbations. These results also suggest that improved camera estimation could further enhance performance.
>
> **Table 5. Impact of various geometry noise.**
> | Method           | ScanQA (CIDEr) |
> |------------------|----------------|
> |AKS | 99.0|
> |DivPrune|99.1|
> |FLoC|99.4|
> | KeyVT + 1% noise  | 100.1 |
> | KeyVT + 5% noise  | 100.0  |
> | KeyVT + 10% noise | 98.4 |
> |------|---|
> | True camera      | 100.7          |
> | VGGT camera      | 100.0          |
>
>
> >Q4: Why “Zero-Shot” in the title?
>
> We highlight “zero-shot” because KeyVT is entirely tuning-free and can be directly applied without requiring training data or example pairs. We will consider refining the title for clarity in the revision.

---

> > ### Author Rebuttal · Reviewer_LddP · 2026-04-03
> >
> > I'd like to thank the authors for the rebuttal;
> > My concerns have been adequately addressed.
> >
> > I'll keep my score for now, and reconsider once the discussions regarding the other reviewers' concerns have finished.

---

### Official Review · Reviewer_hFCv · 2026-03-12

**Soundness:** 3
**Presentation:** 2
**Significance:** 2
**Originality:** 2
**Overall Recommendation:** 4
**Confidence:** 3

**Summary:**

The paper proposes **KeyVT**, a tuning-free hierarchical context selection method for zero-shot 3D question answering. It first selects geometry-aware key views and then compresses tokens across those views using an optimal-transport-based procedure. The method is evaluated on different benchmarks with multiple 2D VLM backbones, and the paper reports consistent improvements over several tuning-free baselines, with performance sometimes approaching training-based methods.

**Compliance With Llm Reviewing Policy:**

Affirmed.

**Final Justification:**

The rebuttal has resolved my concerns with additional experiments and ablations.

**Key Questions For Authors:**

1. The gains over prior tuning-free baselines appear relatively modest, and some metrics show slight drops. Can the authors better contextualize the practical significance of these improvements? For example, are there specific question types or scene settings where KeyVT provides clearer advantages?

2. Can the authors provide additional ablations that pair **KeyT** with alternative view selection strategies, such as AKS or uniform sampling? This would help clarify whether the gains come from KeyT itself or from its interaction with KeyV.

3. How sensitive is the method to the quality of camera geometry, especially on VSI-Bench where poses are estimated rather than given? An analysis with noisy or perturbed poses would strengthen the paper.

4. Are there particular failure cases where KeyVT performs worse than simpler baselines, such as questions requiring evidence spread across many views or scenes with inaccurate geometry? A short failure-case discussion would improve the paper.

**Limitations:**

No. The paper should discuss its limitations more explicitly, especially the modest and non-uniform gains across metrics, the partial disentanglement between KeyV and KeyT, and the dependence on camera geometry quality. It would also be helpful to discuss potential failure cases when geometry is noisy or when relevant evidence is distributed across many views.

**Strengths And Weaknesses:**

### Strengths

- The paper addresses a relevant problem: preserving 3D evidence under a strict visual token budget when using 2D VLMs for 3D-QA. The hierarchical “select views, then compress tokens” design is intuitive and practically motivated.

- The empirical evaluation is reasonably broad, covering three datasets and multiple backbones. The main results show consistent gains over tuning-free baselines, and the ablations suggest that both the KeyV and KeyT stages contribute.

- The presentation is clear, and the method is easy to follow. The qualitative examples help illustrate the intended behavior.

### Weaknesses

- The gains are modest and not uniformly better across metrics: KeyVT usually improves over AKS/FLoC/DivPrune, but often by small margins, while there are also metric drops. Therefore, the claim of “significant improvements” feels somewhat overstated relative to the actual margins.

- The ablations only partially disentangle the two stages. While the paper fairly compares token compression methods under the same KeyV-selected views, it does not test KeyT under alternative view selection strategies. As a result, it is still unclear whether the gains come from KeyT as a generally stronger token compressor or from its synergy with the proposed KeyV stage.

- Since the proposed view selection critically depends on camera geometry, it is important to quantify robustness when geometry is missing or noisy. This is particularly relevant on VSI-Bench, where the camera parameters are estimated rather than provided, yet the paper does not analyze the effect of pose estimation errors on performance.

---

> ### Author Rebuttal · Authors · 2026-03-31
>
> We thank Reviewer hFCv for your thoughtful feedback and constructive suggestions! We appreciate your positive reviews on clear motivation and effective designs. Below are our detailed responses to your remaining questions.
>
> >W1: The gains are modest and not uniformly better across metrics.
>
> - On one hand, we acknowledge that the gains of KeyVT are relatively modest. However, this is not unique to our method and is also observed in prior baselines. For example, on ScanQA, CIDEr scores with LLaVA-OV-7B (AKS: 90.2, DivPrune: 89.5, FLoC: 91.1) and LLaVA-Video-7B (AKS: 99.0, DivPrune: 99.1, FLoC: 99.4) are similarly close. We attribute this to the tuning-free setting, where MLLMs are fixed and only inputs differ, inherently limiting the performance upper bound.
>
> **Table1: Average Rank of baselines. Lower is better.**
> | Dataset    | AKS  | DivPrune | FLoC | Ours |
> |---|--|--|---|--|
> | ScanQA (15 experiments)    | 3.07 | 3.20| 2.13 | 1.27 |
> | SQA3D (3 experiments) | 2.67 | 3.67     | 2.67 | 1.00 |
> | VSI-BENCH (27 experiments)  | 2.78 | 2.67     | 2.70 | 1.85 |
>
> **Table2: p-value on the VSI-BENCH benchmark.**
> | Dataset    | p-value |
> |---|--|
> | KeyVT vs AKS     | $8.0×10^{-8}$|
> | KeyVT vs DivPrune | $5.3×10^{-7}$ |
> | KeyVT vs FLoC  | $9.1×10^{-4}$ |
>
> - On the other hand, to better assess improvements, we further report the average rank and statistical significance. As shown in Table 1, our KeyVT consistently achieves the best (lowest) average rank across all datasets. Moreover, Table 2 presents the p-values on the VIS-BENCH benchmark, demonstrating that KeyVT achieves statistically significant improvements over all baselines (p < 0.001).
>
> >W2: The ablations on KeyT.
>
> **Table3. Ablations on KeyT.**
>
> | Method              | ScanQA (CIDEr) | SQA3D (EM@1) | VSI-Bench (Avg) |
> |----|---|---|---|
> | Uniform  | 93.4           | 54.7         | 32.2            |
> | Uniform + KeyT       | 94.0           | 55.2         | 32.7            |
> | Retrieval           | 97.3           | 57.4         | 32.8            |
> | Retrieval + KeyT    | 98.2           | 57.8         | 33.5            |
> | AKS                 | 99.0           | 57.3         | 33.2            |
> | AKS + KeyT          | 99.8   | 57.7        | 33.9  |
> | KeyV                | 99.5           | 57.6         | 33.2            |
> | KeyV + KeyT         | 100.7          | 57.9         | 34.6            |
>
> Following the suggestion, we report KeyT ablations in Table 3. The results, together with comparisons in the main paper (Table 4), show that KeyT is an effective token compression module that consistently improves various view selection methods.
>
> >W3:  it is important to quantify robustness when geometry is missing or noisy.
>
> **Table 4. Impact of various geometry noise.**
>
> | Method           | ScanQA (CIDEr) |
> |------------------|----------------|
> |AKS | 99.0|
> |DivPrune|99.1|
> |FLoC|99.4|
> | KeyVT + 1% noise  | 100.1 |
> | KeyVT + 5% noise  | 100.0  |
> | KeyVT + 10% noise | 98.4 |
> |------|---|
> | True camera      | 100.7          |
> | VGGT camera      | 100.0          |
>
> Following your advice, we evaluate KeyVT under different levels of camera noise using: $x_{noise} = x_{True} + Uniform(-|x| * \epsilon, |x| * \epsilon $, where $\epsilon$ denotes the noise level (such as 1%, 5% and 10%). As shown in Table 4, KeyVT remains robust and outperforms baselines even under 5% noise, demonstrating strong tolerance to geometry perturbations.
>
> >Q1: The gains over prior tuning-free baselines appear relatively modest.
>
> Please refer to W1 for a more detailed discussion. Empirically, we do not find that the success of KeyVT is limited to specific scenarios. We attribute its performance gains to two key factors: (1) KeyV effectively selects the most relevant sub-scenes conditioned on the query, and (2) KeyT maximizes the amount of useful information fed into the MLLMs under a fixed input budget.
>
> >Q2: Can the authors provide additional ablations on KeyT?
>
> Addressed in W2. We will include these results in the revision.
>
> >Q3: How sensitive is the method to the quality of camera geometry?
>
> We have added additional results and discussion in W3.  These findings suggest that our approach could achieve even stronger performance on VIS-BENCH when paired with more accurate camera estimation models.
>
> >Q4: Are there particular failure cases?
>
> We have included failure cases in the appendix. KeyVT may fail when (1) queries lack sufficient textual guidance (a common limitation of retrieval-based methods), or (2) camera geometry is inaccurate. In practice, VGGT provides reasonably accurate estimates in most cases. We will add more visualizations and discussion in the revision. Regarding scenarios where questions require multiple views, we find that KeyVT still outperforms the baselines. This advantage mainly stems from KeyVT’s ability to compress redundant tokens across views, thereby allowing more informative evidence to be fed into the MLLMs under a fixed input budget.

---

> > ### Author Rebuttal · Reviewer_hFCv · 2026-04-03
> >
> > Thanks for the rebuttal. It has resolved my concern.

---

### Decision · Program_Chairs · 2026-04-30

**Decision:**

Accept (regular)

**Comment:**

The paper proposes KeyVT, a tuning-free hierarchical context selection method for zero-shot 3D question answering. It operates under strict visual token budgets by selecting geometry-aware key views (KeyV) and subsequently compressing tokens using an optimal-transport-based procedure (KeyT).

The panel recognized the practical motivation of the problem and the intuitive nature of the hierarchical design, but raised several concerns in the initial reviews. The authors provided a rigorous and heavily empirical rebuttal that satisfied the strict requirements of the panel. They supplied statistical significance (p-values) to contextualize their performance improvements, provided thorough ablations that effectively isolated the contributions of KeyV and KeyT, and added experiments demonstrating robust performance even under 5% injected camera noise. Furthermore, the authors articulated a strong geometric justification for OT, demonstrating that it prevents the solution from collapsing onto a few dominant regions compared to standard k-means.

While the overarching contribution is heavily weighted toward a context-selection pipeline rather than a fundamentally new 3D reasoning paradigm, the methodology is exceptionally well-validated. The authors diligently answered every technical critique. Given the solid technical execution and the positive post-rebuttal consensus, the paper meets the threshold for acceptance.